# MotionDreamer: one-to-many Motion Synthesis with Localized Generative Masked Transformer

**Yilin Wang**[1]**, Chuan Guo**[1]**, Yuxuan Mu**[3]**, Muhammad Gohar Javed**[1]**, Xinxin Zuo**[2]**,
Juwei Lu**[4]**, Hai Jiang**[1]**, Li Cheng**[1]
[1]University of Alberta     [2]Concordia University     [3]Simon Fraser University
[4]Noah's Ark Lab, Huawei Canada

## ABSTRACT

Generative masked transformers have demonstrated remarkable success across various content generation tasks, primarily due to their ability to effectively model large-scale dataset distributions with high consistency. However, in the animation domain, large datasets are not always available. Applying generative masked modeling to generate diverse instances from a single MoCap reference may lead to overfitting, a challenge that remains unexplored. In this work, we present Motion-Dreamer, a localized masked modeling paradigm designed to learn internal motion patterns from a given motion with arbitrary topology and duration. By embedding the given motion into quantized tokens with a novel distribution regularization method, MotionDreamer constructs a robust and informative codebook for local motion patterns. Moreover, a sliding window local attention is introduced in our masked transformer, enabling the generation of natural yet diverse animations that closely resemble the reference motion patterns. As demonstrated through comprehensive experiments, MotionDreamer outperforms the state-of-the-art methods that are typically GAN or Diffusion-based in both faithfulness and diversity. Thanks to the consistency and robustness of the quantization-based approach, MotionDreamer can also effectively perform downstream tasks such as temporal motion editing, crowd animation, and beat-aligned dance generation, all using a single reference motion. Visit our project page: https://motiondreamer.github.io/.

## 1 INTRODUCTION

Motions could be roughly interpreted as coherent and natural compositions of finite internal patterns. For example, a *breaking* can be performed by a freestyle composition of breaking dance skills, such as *baby freeze*, *helicopter*, *kick up*, *back spin*, etc. Learning these internal patterns from a single reference motion allows for the generation of diverse yet consistent motions that closely resemble the reference. This is particularly useful when data is scarce (e.g., in the case of animal motion) or when the content needs to be constrained. Existing works (Li et al., 2022a; Raab et al., 2024; Li et al., 2023b) have attempted to address this using GAN (Goodfellow et al., 2014; Denton et al., 2015; Zhang et al., 2017) or Diffusion Model (Ho et al., 2020; Rombach et al., 2022; Tevet et al., 2023), where the distribution of internal patterns is modeled implicitly, or through motion matching, which matches and blends exemplar internal patterns without learning latent embedding. However, these approaches either struggle with limited expressiveness of internal patterns or fail to achieve diverse and natural synthesis.

Recent advances in motion synthesis have witnessed great success in the use of generative masked transformers (Zhang et al., 2023a; Jiang et al., 2023; Guo et al., 2024; Pinyoanuntapong et al., 2024), largely due to their efficient embeddings and explicit distribution modeling. These learned models quantize motion into tokens and represent them with an explicit categorical distribution in the discrete space defined by the codebook. It is done by randomly masking and predicting the specific masked tokens in context. However, they usually rely on large-scale datasets, applying global self-attention to learn the in-context distribution. When adapted to single or few sequences, these models tend to overfit to sequence-wise global patterns instead of generating novel sequences based on the distribution of internal patterns. This results in model collapses, since the standard transformer layers attend all the tokens in a sequence for positional encoding and self-attention.

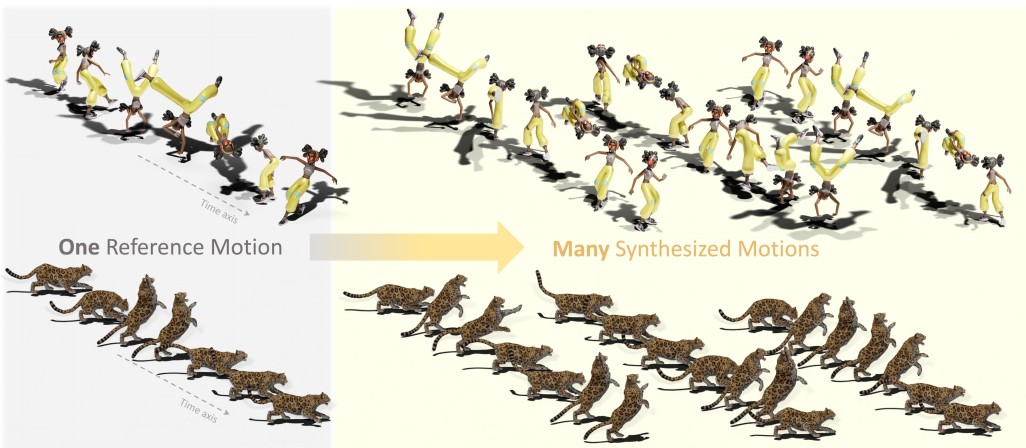

Figure 1: Overview of the one-to-many motion synthesis. A single reference motion with arbitrary skeletons can be applied to generate natural and diverse novel motions while preserving the reference local motion patterns. Above shows the diverse generations from MotionDreamer of a girl doing breakdance (upper); a jaguar attacking (bottom).

In this paper, we propose MotionDreamer, a generative masked transformer specifically designed for one-to-many motion synthesis, as presented in Figure 1. The key idea is to learn the explicit categorical distribution of the internal patterns by strategically narrowing the receptive field of transformer layers, as illustrated in Figure 2. This is achieved by our proposed localized generative masked transformer (Local-M), which incorporates a carefully designed sliding window local attention layer (SlidAttn) as the backbone. This structure focuses on capturing local dependencies of quantized motion tokens. The SlidAttn layer divides the input motion token sequence into overlapping local windows, where motion tokens are attended to using window-wise relative positional encoding (Shaw et al., 2018). The resulting local attention scores are then aggregated through overlap attention fusion, to better preserve cross-window coherence. During training, a quantized codebook is first optimized to embed internal patterns of a single reference motion. To prevent the under-utilization of code entries, a codebook distribution regularization technique is further introduced. Once the reference motion is represented with motion tokens from the codebook, Local-M is trained to model the explicit categorical distribution of internal patterns through generative masked modeling, where varying masked portions of motion tokens are progressively predicted. A differentiable dequantization strategy is also integrated to facilitate the optimization in motion space, by applying the sparsemax activation function (Martins & Astudillo, 2016) instead of the traditional argmax operation.

To summarize our main contributions: (1) We introduce MotionDreamer, a novel localized generative masked modeling paradigm for motion synthesis based on single reference motion. With just one training motion sequence, it effectively addresses the key issues: a) codebook collapse, by promoting a highly utilized codebook through a distribution regularization technique, and b) overfitting, by shrinking the receptive field of transformer using the proposed sliding window local attention layer as the backbone. (2) Our MotionDreamer faithfully preserves the internal patterns of reference motion while notably diversifying the synthesized motions. It achieves state-of-the-art comprehensive performance improving the harmonic mean by 19%, and earning the highest score in perceptual assessments. (3) MotionDreamer is also shown to work well in downstream applications of temporal editing, crowd animation and beat-aligned dance synthesis, by leveraging only a single reference motion. Our project page includes visualization demos and implementation codes.

## 2 RELATED WORKS

### 2.1 SINGLE INSTANCE SYNTHESIS

Single reference based motion synthesis presents as the extension of single instance learning in the image domain (Rott Shaham et al., 2019; Zhang et al., 2021; Hinz et al., 2021; Granot et al., 2022) to motion domain. In image synthesis field, Rott Shaham et al. (2019) generates diverse yet reminiscent results from a single image based on generative adversarial learning through the hierarchical

image pyramid framework. On the other hand, Granot et al. (2022) argues that utilizing patch-based nearest neighbor matching results in a more faithful and robust synthesis based on single image. In the motion synthesis domain, Ganimator (Li et al., 2022a) adapts the hierarchical GAN architecture from Rott Shaham et al. (2019) to learn and generate diverse motion sequences from a single motion instance. SinMDM (Raab et al., 2024) approaches the hierarchical generation process by the diffusion model based on a light-weight UNet structure with local attention layers (Arar et al., 2022). Both methods model the internal patterns of the single motion implicitly in a continuous latent space, which potentially induces more out-of-distribution during synthesis, leading to limited capability of representing internal patterns and unnatural artifacts in generated motions. Inspired by Granot et al. (2022), GenMM (Li et al., 2023b) generates novel motions through multi-level matching and blending the reference motion patches simply based on nearest neighbour search. However, the diversity of GenMM (Li et al., 2023b) is limited as it basically shuffles and merges original motion patches from reference motion without learning the underlying distribution, and artifacts are more frequently observed in short and highly dynamic sequences. In this work, we propose a novel localized generative masked transformer for this task, which presents an effective learning paradigm with strong expressiveness of local patterns as well as natural and diverse generations. We provide more insights about the mechanism in Section 3 and potentials in empirical results in Section 4.

## 2.2 MOTION GENERATIVE TRANSFORMERS

Inspired by the impressive success of generative transformers in image synthesis (Ramesh et al., 2021; Chang et al., 2022; Li et al., 2023a; Yu et al., 2023), several recent works (Zhang et al., 2023a; Jiang et al., 2023; Lu et al., 2024; Guo et al., 2024; Pinyoanuntapong et al., 2024) adapt the generative transformer framework to approach text-driven human motion synthesis. Zhang et al. (2023a) establishes a simple CNN-based VQ-VAE (van den Oord et al., 2017) for transforming human motion sequences into discrete motion token sequences, and a modified generative pretrained transformer(GPT) backbone is utilized for learning to generate novel human motions conditioned on text prompt. Jiang et al. (2023) learns mixed motion tokens and text tokens simultaneously utilizing language model backbone.Lu et al. (2024) leverages hierarchical VQ-VAE Razavi et al. (2019) and a hierarchical GPT for more precise whole-body motion generation. Guo et al. (2024) and Pinyoanuntapong et al. (2024) both adapt generative masked modeling to optimizing the transformer for enhanced semantics mapping between text and motion, where a scheduled varying portion of tokens are masked for predictions during training iterations. Inspired by the effectiveness of explicit distribution modeling for motion sequences shown in these works, our method employs the generative masked transformer for MotionDreamer and specifically adapts the framework to single instance based motion synthesis by shrinking the receptive field of transformer layers. With our proposed MotionDreamer, we demonstrate strong capability of capturing fine-grained local motion features, and diversifying their combinations with plausible transitions for novel motions synthesis.

## 3 LOCALIZED MOTION GENERATIVE TRANSFORMER

Given a single reference motion $m_{1:L}$ of length $L$ and arbitrary skeleton topology, the goal is to generate a generally novel motion sequence $\tilde{m}_{1:L_g}$ of arbitrary length $L_g$ that preserves the skeleton structure and underlying local patterns of the reference motion. As illustrated in Fig 2 (a) and (b), we establish a localized generative masked transformer to model the explicit categorical distribution of internal patterns on a learned discrete latent space $\mathcal{C}$ (namely *codebook*), from which we sample a tuple of embeddings $c_{1:N_g}$(namely *motion tokens*) in an auto-regressive manner to generate novel motion sequences. Our method incorporates two core components: codebook distribution regularization and Sliding Window Local Attention (SlidAttn). The codebook regularization mitigates the codebook collapse when training a single sequence, promoting more uniform and diverse use of code entries during tokenization, while SlidAttn ensures effective modeling of local transitions by operating tokens within overlapping windows.

## 3.1 SINGLE MOTION TOKENIZATION

The tokenization process encodes and maps the local motion segments in the original motion space $\mathbf{M}$ into motion tokens in a quantized discrete latent space $\mathbf{C}$. During training, a finite codebook is preset to represent the motion token space: $\mathcal{C} = \{c_i\}_{i=1}^K \subset \mathbb{R}^d$, where $K$ is the number of

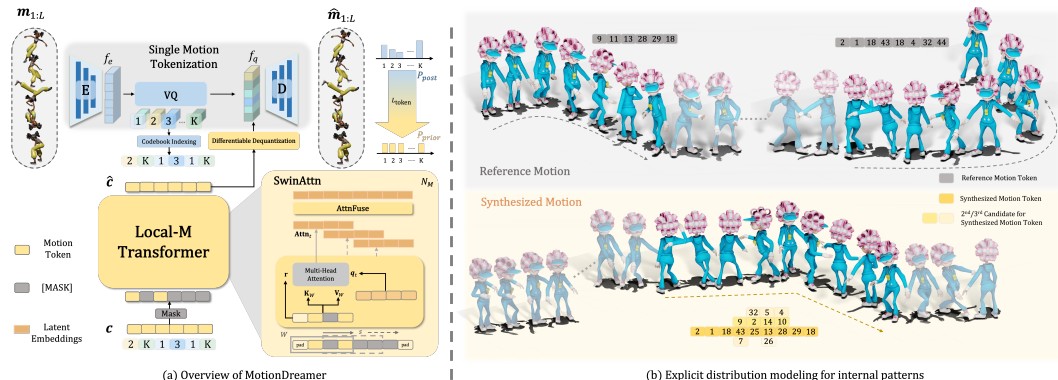

(a) Overview of MotionDreamer        (b) Explicit distribution modeling for internal patterns

Figure 2: (a) Overview of MotionDreamer based on localized generative masked transformer. The single reference motion $m_{1:L}$ is embedded as motion tokens $c$ by optimizing a codebook through vector quantization, where a codebook distribution regularization loss $\mathcal{L}_{token}$ is additionally introduced. The Local-M transformer learns the local dependencies of motion tokens through sliding window local attention (SlidAttn) layers. The SlidAttn layer attends tokens within each unfolded overlapping window for attention based on learnable query and relative positional embeddings. Attention outputs are merged through overlap attention fusion (AttnFuse). (b) Visualization of the explicit distribution modeling for internal patterns. MotionDreamer learns to express and diversify the combination of internal patterns with explicit categorical distribution of motion tokens, which is visualized as multiple token candidates predicted by Local-M given previous generated ones.

code entries. The reference motion $m_{1:L}$ is encoded as a sequence of feature vectors through 1D convolutional encoder $E$:

$$z_{1:N} = \{z_1, z_2, \ldots, z_N\} = E(m_{1:L}) \tag{1}$$

where $N = \frac{L}{h}$, and $h$ is the down-sampling factor of the encoder. Through vector quantization (VQ) (van den Oord et al., 2017), noted as $Q(\cdot)$, each of the feature vectors is mapped to the nearest code entry in codebook $\mathcal{C}$, resulting in a sequence of motion tokens:

$$c_{1:N} = \{c_1, c_2, \ldots, c_N\} = Q(z_{1:N}; \mathcal{C}). \tag{2}$$

Finally, the motion token sequence is projected back to original motion space $m$ through 1D convolutional decoder $D$ as the reconstructed motion sequence:

$$\hat{m}_{1:L} = D(c_{1:N}). \tag{3}$$

### 3.1.1 CODEBOOK DISTRIBUTION REGULARIZATION

Due to the constraint of highly limited data and the imbalanced temporal distribution of internal patterns, training vector quantizer on single motion sequence is more likely to struggle with issues such as codebook collapse. Although common strategies such as exponential moving average (EMA) and codebook reset (Razavi et al., 2019; Zhang et al., 2023a; Guo et al., 2024) result in good reconstruction at the training phase, we spot that the motion patterns are still often lost or blurred with unnatural transition between poses at the generation phase, shown in Figure 5. This could be ascribed to the under-utilization of code entries, which may introduce invalid tokens that disturb the expression of learned patterns. In order to mitigate this, we propose to minimize the KL divergence between predicted token distribution and a pre-assumed prior token distribution (Zhang et al., 2023b) to encourage a more uniform distribution of the effective code entries:

$$\mathcal{L}_{token} = KL(P_{post}, P_{prior}) = -\sum_{k=1}^{K} p_k \log(\frac{1/K}{p_k}) \tag{4}$$

where $p_k$ is the posterior distribution of code entries approximated by the average of all the quantized one-hot vectors indicating the selected code entries. The overall optimization objective for training the single motion quantizer is then established as:

$$\mathcal{L}_{VQ} = \mathcal{L}_{rec} + \beta_q \mathcal{L}_q + \beta_k \mathcal{L}_{token} \tag{5}$$

where $\mathcal{L}_{\text{rec}} = ||\boldsymbol{m} - \hat{\boldsymbol{m}}||_1$ is the motion reconstruction loss, and $\mathcal{L}_q = \sum_{i=1}^{N} ||f_e - \text{sg}(f_q)||^2$ is the standard VQ commitment loss ($\text{sg}(\cdot)$ denotes the stop gradient operation) (van den Oord et al., 2017). Broader impacts of $\mathcal{L}_{\text{token}}$ on other VQ-based motion representation methods are discussed in Appendix A.7.

## 3.2 LOCAL-M TRANSFORMER

Local-M Transformer $p_\phi$ models the explicit distribution of internal patterns through progressively predicting randomly masked motion tokens. Specifically, a varying fraction $r$ of motion tokens $\boldsymbol{c}_{1:N}$ are masked out in iteration timestep, replaced with a special [MASK] token. The masking ratio is obtained through a cosine scheduling function $r = \gamma_m(\mu) = \cos(\frac{\pi\mu}{2}) \in [0, 1]$, where $\mu \sim \mathcal{U}(0, 1)$ is randomly sampled, following Chang et al. (2022). Our transformer $p_\phi$ is optimized by minimizing the negative log-likelihood of the target predictions:

$$\mathcal{L}_{\text{mask}} = \sum_{\boldsymbol{c}_r^m = [\text{MASK}]} - \log p_\phi(\boldsymbol{c}_r^m | \boldsymbol{c}^m) \tag{6}$$

where $\boldsymbol{c}^m$ is the token sequence after masking. To facilitate the optimization by directly minimizing the motion reconstruction loss in addition to the masked modeling loss, we integrate a differentiable dequantization strategy to enable gradient flow from decoded motion to Local-M transformer. By incorporating sparsemax$(\cdot)$ activation function (Martins & Astudillo, 2016), the module simulates the argmax selection of motion tokens in a differentiable manner.

The overall loss function of Local-M transformer is:

$$\mathcal{L}_{\text{M}} = \mathcal{L}_{\text{mask}} + \lambda_{\text{rec}} \mathcal{L}_{\text{rec}} \tag{7}$$

where $\lambda_{\text{rec}}$ is the hyper-parameter that balance the reconstruction loss with mask modeling loss. $\mathcal{L}_{\text{rec}} = ||\boldsymbol{m} - \hat{\boldsymbol{m}}||_1$ is computed where $\hat{\boldsymbol{m}}$ is the motion decoded from the predicted token sequence.

### 3.2.1 SLIDING WINDOW LOCAL ATTENTION

In our problem setting, naively optimizing standard transformer blocks with $\mathcal{L}_{\text{mask}}$ leads to severe overfitting issue as the model fails to capture local dependencies of motion tokens given a single sequence. To address this problem, we introduce the SlidAttn layer. Each SlidAttn layer unfolds the motion token sequence into overlapping local windows, and computes attention within each local window, the output of which is aggregated through overlap attention fusion (AttnFuse). With SlidAttn, $p_\phi$ approximates the joint distribution of motion tokens by modeling the local transitions in sliding windows.

**SlidAttn Mechanism**  By unfolding the input motion token sequence $\boldsymbol{c}_{1:N}$ with stride $S$ into overlapping local windows of size $2W + 1$, the key matrices $\mathbf{K}_{t-W:t+W}$ and value matrices $\mathbf{V}_{t-W:t+W}$ in each window can be obtained, where $t = \{W, W + S, W + 2S, \dots, W + BS\}$. Note that in our case, having the windows overlapped yet keeping a small overlap length is crucial for learning smooth transitions across windows on top of local patterns within windows while preserving typical internal patterns. In this paradigm, utilizing absolute positional encoding (Vaswani et al., 2017) within local windows may induce severe boundary artifacts especially with only a single sequence. To mitigate the issue, we employ window-wise relative positional encoding $\mathbf{r} = \text{RelPos}(W)$ adapted from Shaw et al. (2018) which ensures the attention mechanism to focus on the relative distances instead of absolute positions. Furthermore, we introduce learnable query $\mathbf{q}_t$ which allows the model to adaptively attend to important tokens across different windows. This encourages the model to better capture the fine-grained transitions and internal patterns that vary between different types of motions. The attention computation shown in Figure 2 attending tokens within each local window can be formulated as:

$$\mathbf{Attn}_t = \text{softmax}\left(\frac{\mathbf{q}_t \mathbf{K}_W + \mathbf{r}}{\sqrt{d_k}}\right) \mathbf{V}_W \tag{8}$$

where $\mathbf{K}_W$ and $\mathbf{V}_W$ are the abbreviation for $\mathbf{K}_{t-W:t+W}$ and $\mathbf{V}_{t-W:t+W}$, respectively.

**AttnFuse**  Standard average pooling (Vaswani et al., 2017; Beltagy et al., 2020) across the overlapping windows may require large amount of padding for the input token sequence in order to ensure consistent input-output sequence length, inducing indispensable artifacts. To address this, AttnFuse

aligns the attention output $\mathbf{Attn}_t$ of each window as same as the way the input sequence is unfolded with trivial padding (with padding size $\leq$ window size $\ll L/h$), and blends the overlapped regions with average voting while straightly passing through the others, resulting in the final aggregated output. This not only mitigates the padding artifacts that noise the distribution of internal patterns, but also allows for preserving better coherence for generating smooth motions.

### 3.2.2 INFERENCE PROCESS

At the inference stage, we synthesize the novel motion $\tilde{m}_{1:L_g}$ of arbitrary length $L_g$ using Local-M in a sliding window based auto-regressive manner, by progressively filling in a fully-masked template token sequence. The details of the inference process are elaborated in Appendix A.3.

## 4 EXPERIMENT AND RESULTS

### 4.1 IMPLEMENTATION DETAILS

**Dataset**    We collect 30 motion sequences from Mixamo (Mixamo, 2023) with human-like skeleton and 30 motion sequences with skeletons of animal and artist-crafted creatures from Truebone-ZOO (Studio, 2023) dataset to form the *SinMotion* dataset used for evaluation. There are 30 long sequences ($> 600$ frames) and 30 short sequences ($\leq 600$ frames).

**Evaluation Metrics**    We apply 5 sets of metrics to measure the expressiveness of local motion patterns and synthesis diversity respectively following Li et al. (2022a) and Raab et al. (2024): (1) *Coverage(%)* measures the percentage of local motion patterns in reference motions that are presented in generated motions, which indicates the faithfulness or expressiveness of the internal patterns. (2) *Global Diversity* quantifies the overall variation in the generated motion sequences based on patched nearest neighbor (Li et al., 2022a). It assesses the ability of the synthesis method to produce a wide range of distinct motions that differ significantly from each other. (3) *Local Diversity* evaluates local frames diversity. (4) *inter diversity* measures the diversity between synthesized motions by averaging per-frame distances. (5) *intra diversity diff* measures the local pattern distribution similarity between the reference motion and generated motions, where the statistic of the pattern distribution is approximated by internal sub-window distances. Computation details for (1), (2), (3) are presented in Appendix A.5.

As the evaluation results are comprehensively assessed by lots of metrics for different aspects, we also introduced *Harmonic Mean* following Raab et al. (2024) to combine the effects of all the metrics. The Harmonic Mean is established as early as in van Rijsbergen (1979); Chinchor (1992), providing a more robust comparison in cases where metric values vary in a large range, and has been widely applied in many machine learning fields (Powers, 2011; Taha & Hanbury, 2015; Chicco & Jurman, 2020). It is formulated as $HE = H/(\frac{1}{x_1} + \frac{1}{x_2} + \cdots + \frac{1}{x_H})$, where $H$ is the number of assessed metrics and $x_i$ is the standardized score of the $i$-th metric. Given the nature of this task, we give the highest weight to metric (1) and lower weights to (2)-(5).

For implementation details of the model architecture, parameter settings and the parameter tuning ablations, please refer to Appendix A.2 and Appendix A.6.

### 4.2 COMPARISON WITH STATE-OF-THE-ART METHODS

We compare our method with all the existing approaches for single reference motion synthesis on the collected SinMotion dataset. The compared framework includes GAN (Li et al., 2022a), diffusion model (Raab et al., 2024) and non-parametric optimization method (Li et al., 2023b).

**Quantitative Comparison**    For each reference motion in the SinMotion dataset, we randomly generate 20 samples with $L_g = L$ for measuring the metrics. Table 1 presents the quantitative results of our method compared to the state-of-the-art methods. Our method achieves state-of-the-art results on individual metrics (1)-(5), and the best results in the comprehensive metrics Harmonic Mean. GenMM (Li et al., 2023b) reaches higher coverage but with limited diversity. SinMDM (Raab et al., 2024) presents good diversity and competitive overall results, while perceptual quality is traded off as typical artifacts can be frequently observed in the generated motions, illustrated in Figure 3 and video demos in supplementary materials.

Table 1: Quantitative comparison with state-of-the-art methods on SinMotion dataset. **Bold** marks the best result, and underline notes the second best.

| Method | Coverage (%) ↑ | Global Div. ↑ | Local Div. ↑ | Inter Div. ↑ | Intra Div. ↓ | Harmonic Mean ↑ |
|---|---|---|---|---|---|---|
| Ganimator (Li et al., 2022a) | 91.27 | 0.97 | 0.90 | 0.20 | 0.34 | 0.26 |
| GenMM (Li et al., 2023b) | **95.29** | 0.50 | 0.49 | 0.19 | 0.29 | 0.32 |
| SinMDM (Raab et al., 2024) | 91.82 | 1.31 | **1.20** | 0.22 | 0.32 | 0.36 |
| Ours | 93.47 | **1.33** | 1.17 | **0.25** | **0.28** | **0.43** |

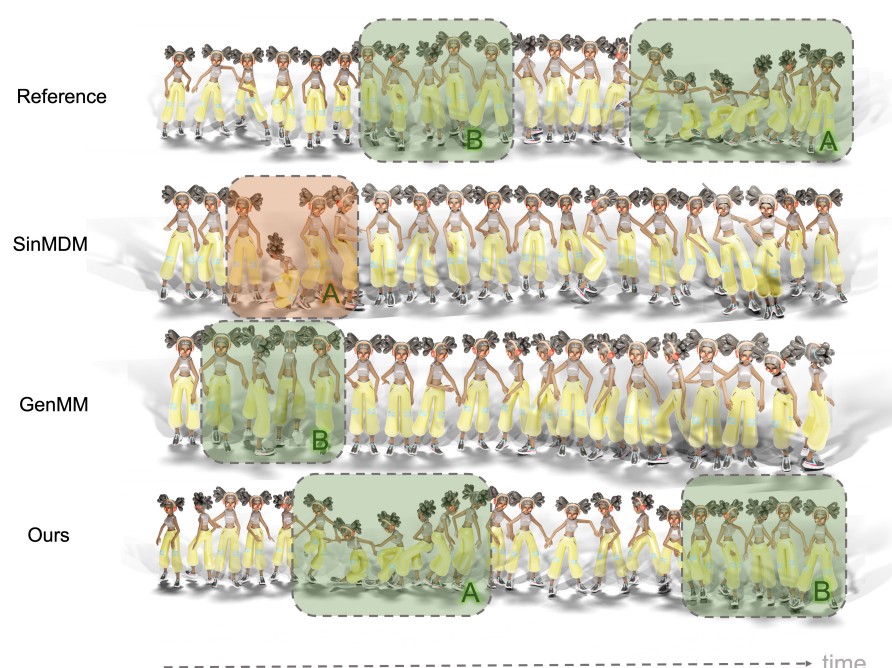

Figure 3: Qualitative comparison on "hiphop dance" sample from Mixamo. Pattern A and B refer to two difficult patterns presented in the reference motion. Patterns that show up in generated motions are framed out marked as either success or failure according to its quality.

**Qualitative Comparison** We select the top 2 difficult yet typical patterns in the "hiphop dance" sample by integrating opinions of 5 dancers, and compare the visualization results of generated motions from our method, SinMDM(Raab et al., 2024) and GenMM (Li et al., 2023b) by constraining $L_g = L$ in Figure 3. As visualized, the generation of both SinMDM (Raab et al., 2024) and GenMM (Li et al., 2023b) tend to lose track of pattern A, which is a speedy and complex sub-part of a person squatting down while spinning. GenMM (Li et al., 2023b) fails to present pattern A in the generated sample, while SinMDM (Raab et al., 2024) generates it with an unnatural transition of a sudden squatting down pose. The generation of our method succeeds to express pattern A and B as well as other local patterns, while getting generally distinct from the reference motion. It is worth mentioning that GenMM achieves higher quantitative results on coverage while the qualitative result reveals its lower fidelity in presenting highly dynamic and complex patterns compared to our method. More generated samples and diversity visualization can be found on our website linked in Appendix A.1.

**User Study** To further assess the perceptual performance, we conduct a user study among 20 participants. For each of the 3 reference motions, we present 3 randomly generated motions of each method. The participants are asked to assess each generated motion by (1) Coverage: it covers all the noticeable patterns in the reference motion; (2) Diversity: it is generally novel and distinguishable from the reference motion; (3) Naturalness: it is plausible and natural with smooth transition. The assessment is based on 5 levels of scores from "1 strongly disagree" to "5 strongly agree" with each of the above statements. Figure 4 presents the score distribution and averaged score of each method in each assessed aspect. As shown in Figure 4, our method is scored highest in coverage and diversity, and is closely comparable with GenMM (Li et al., 2023b) in naturalness. Noted that GenMM (Li et al., 2023b) is a non-parametric method that directly matches and blends the

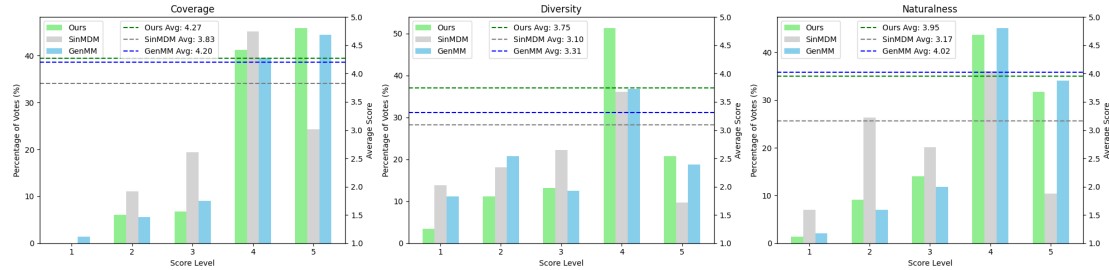

Figure 4: Score distribution and average score results from user study. The score level ranges from 1 to 5 of assessing Coverage, Diversity and Naturalness. The bars align with the right y-axis referring to percentage of votes of each method in each score level, and the horizontal lines align with left y-axis labeling the average score of each method.

Table 2: Ablation study on VQ regulatization strategies on SinMotion dataset. **Bold** text marks the best result.

| Method | VQ Perplexity ↑ | Coverage (%) ↑ | Global Div. ↑ | Local Div. ↑ | Inter Div. ↑ | Intra Div. ↓ | Harmonic Mean ↑ |
|---|---|---|---|---|---|---|---|
| Ours w/o. $\mathcal{L}_{\text{token}}$ | 24.56 | 87.26 | **1.35** | 1.14 | 0.18 | **0.26** | 0.34 |
| Ours | **28.13** | **93.47** | 1.33 | **1.17** | **0.25** | 0.28 | **0.43** |

motion patches from the reference motion without extracting latent representation, which naturally attains good coverage and naturalness in most of the cases. Our method involves generating the motion token sequence in a discrete latent space and mapping back to the motion space, yet still reaches competitive perceptual naturalness as GenMM (Li et al., 2023b). Moreover, our method has greatly surpassed SinMDM (Raab et al., 2024) and GenMM (Li et al., 2023b) in the user perceptual evaluation. It is worth mentioning that while our method gets close diversity measures compared to SinMDM (Raab et al., 2024) quantitatively, our superior perceptual performance is demonstrated in diversity as well as other assessed aspects.

### 4.3 ABLATION: CODEBOOK DISTRIBUTION REGULARIZATION

Mitigating the under-utilization of code entries during the single motion tokenization phase plays a crucial role in preserving the local motion patterns. In this work, we introduce codebook distribution regularization loss $\mathcal{L}_{\text{token}}$ based on KL divergence in addition to the commonly used strategies (EMA and codebook reset) for this purpose. We run the training of single motion tokenization without $\mathcal{L}_{\text{token}}$ and compare the results with our method quantitatively in Table 2. We introduce VQ perplexity to quantify the utilization of code entries, which is formulated as VQ Perplexity $= \exp(-\sum_{k=1}^{K} p_k \log(p_k))$. As shown, the VQ perplexity and coverage drop dramatically without $\mathcal{L}_{\text{token}}$. To further demonstrate it perceptually, we visualize in Figure 5 the qualitative comparison of the expressiveness of local patterns. We select a local window from sample "house dancing", and retrieve the nearest neighbour local window in the generated motions from our method and ours without $\mathcal{L}_{\text{token}}$. The character in Figure 5 is colored according to the per-frame similarity score between the generated window and the reference window, with color closer to green representing higher similarity while color closer to orange referring to lower similarity. As visualized, training with $\mathcal{L}_{\text{token}}$ contributes to a faithful expression of the reference local motion patterns. Otherwise, the motion patterns get blurred with redundant poses and transitions.

Table 3: Ablation study on architecture of Local-M transformer on SinMotion dataset. **Bold** text marks the best result, and underline notes the second best.

| SlidAttn | Diff. dequant | Coverage (%) ↑ | Global Div. ↑ | Local Div. ↑ | Inter Div. ↑ | Intra Div. ↓ | Harmonic Mean ↑ |
|---|---|---|---|---|---|---|---|
| | | 98.82 | 0.03 | 0.04 | 0.02 | 0.08 | 0.07 |
| | ✓ | **99.26** | 0.01 | 0.01 | 0.01 | 0.03 | 0.03 |
| ✓ | | 90.42 | **1.38** | **1.21** | 0.20 | 0.30 | 0.35 |
| ✓ | ✓ | 93.47 | 1.33 | 1.17 | **0.25** | **0.28** | **0.43** |

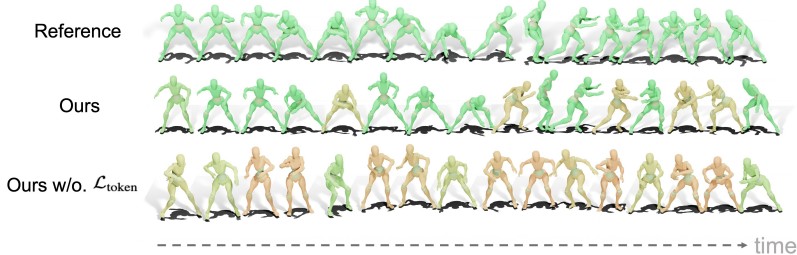

Figure 5: Ablation study on codebook distribution regularization technique based on optimizing $\mathcal{L}_{\text{token}}$. Color closer to green representing higher per-frame similarity while color closer to orange referring to lower similarity.

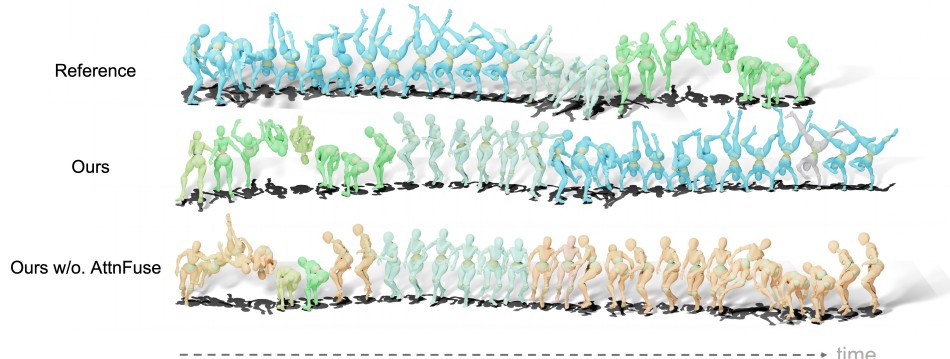

Figure 6: Ablation study on overlap attention fusion (AttnFuse). "Ours w/o. AttnFuse" refers to applying standard average pooling aggregation as the alternative baseline to AttnFuse. "backflip", "handstand" pattern and transition between two patterns are marked. For generated motions, color closer to the pattern colors indicates higher per-frame similarity with the corresponding pattern, while color closer to orange indicates lower similarity.

## 4.4 ABLATION: LOCAL-M TRANSFORMER

We conduct an ablation study on the architecture of Local-M transformer in Table 3. The baseline is backboned on standard transformer block optimized solely by generative masked modeling loss $\mathcal{L}_{\text{mask}}$. As presented in Table 3, the standard transformer backbone fails to learn the local motion token transitions and instead overfits to the single reference motion. By replacing the standard transformer blocks with SlidAttn blocks, the Local-M transformer is capable of generating diverse novel motions, yet with limited expressiveness of internal patterns. The coverage increases with differentiable dequantization enabling stronger constraints directly in motion space, which encourages more precise representation of motions. Integrating differentiable dequantization with SlidAttn during training results in the best overall performance according to the Harmonic Mean.

**Overlap Attention Fusion Effectiveness** The mechanism of SlidAttn is highlighted with the overlap attention fusion as an alternative to the average pooling aggregation of standard local attention layers. In Figure 6, we particularly look into the effects of overlap attention fusion from qualitative results by comparing it to the standard average pooling aggregation which is denoted as "Ours w/o. AttnFuse". We highlight the signature pattern "backflip", "handstand" and transitions between this two patterns. In the generated motions of the two compared methods, the closer the color is to the pattern color, the higher per-frame similarity to the corresponding pattern is indicated, while color closer to orange indicates lower similarity to either pattern. We compare generated motions of similar motion tokens from the two methods as they share the same VQ codebook. As presented in Figure 6, our method with overlap attention fusion achieves faithful expression of the patterns as well as seamless globally distinct re-arrangement of the two patterns by generating natural transitions. Meanwhile, the baseline with average pooling aggregation fails to globally diversify the patterns with unnatural transitions and noisy motion patterns.

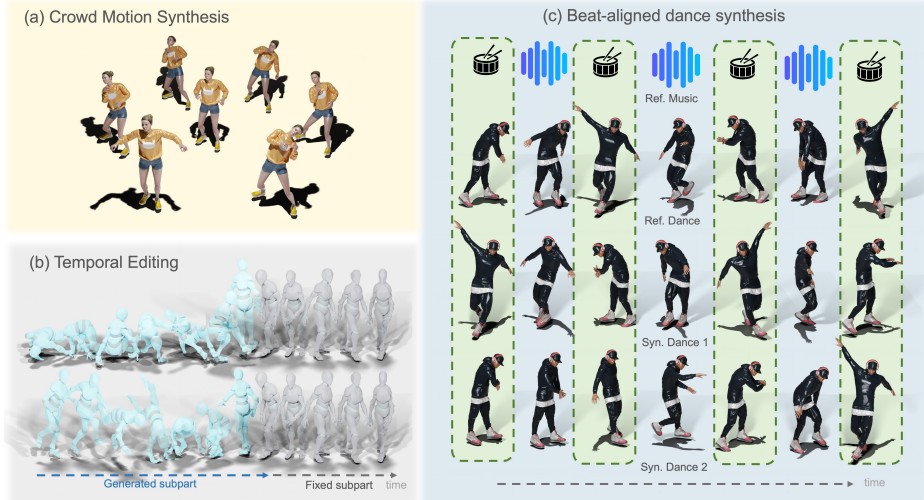

Figure 7: Application visualization. (a) crowd animation: a crowd doing warm-up. (b) Temporal Editing: floorcombo with the ending part fixed, different internal patterns generated. (c) Beat-aligned Dance Synthesis: the green boxes marks the keyframes align with the music beats.

## 4.5 APPLICATION

**Crowd Animation** The unconditional generation of the one-to-many motion synthesis can efficiently provide a diverse set of characters with similar structure performing the same internal patterns in different ways, as shown in Figure 7(a).

**Temporal Editing** The goal of temporal editing is to edit the reference motion by re-generating an arbitrary subpart with the learned local patterns, which can be realized through masking only the template tokens within the assigned editing subpart. As shown in Figure 7(b), MotionDreamer can reproduce novel samples by fixing the end part of reference motion "floorcombo".

**Beat-aligned Dance Synthesis** Another interesting application we explore is a variation from the music-to-dance tasks (Li et al., 2021; 2022b; Tseng et al., 2023), the beat-aligned dance synthesis based on single instance, where the generated motions are ensured to preserve the keyframe poses on the music beats provided with the reference motion. To accommodate this task, we incorporate the beat features using librosa (McFee et al., 2015) as an auxiliary temporal-aligned feature for the motion tokens by employing an additional pair of light-weight encoder-decoder. As visualized in Figure 7(c), the synthesized dance clip can well align with the typical poses on the beats yet well diversifying them, and the dance patterns are also internally reminiscent yet generally diverse.

For implementation details please refer to Appendix A.4 and for video demos go to Appendix A.1.

## 5 CONCLUSION

In this work, we introduce MotionDreamer, a novel one-to-many motion synthesis framework based on localized generative masked modeling. Our method leverages codebook distribution regularization to address the codebook collapse and under-utilization, ensuring a diverse representation of motion patterns even with a single motion. Furthermore, we propose a sliding window local attention (SlidAttn) mechanism that effectively captures local dependencies and smooth transitions across overlapping windows, significantly improving both the fidelity and diversity of generated motions. Through comprehensive experiments, we demonstrate that MotionDreamer achieves state-of-the-art performance, outperforming existing approaches in generating natural and diverse motions from a single reference, and shows strong potential in practical applications such as temporal editing and beat-aligned dance synthesis. More future directions are discussed in Appendix A.8.

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

# A APPENDIX

## A.1 VIDEO DEMOS

The video version of visualization in the paper as well as demos for more generation results and application results can be found on our project page link:

https://motiondreamer.github.io/.

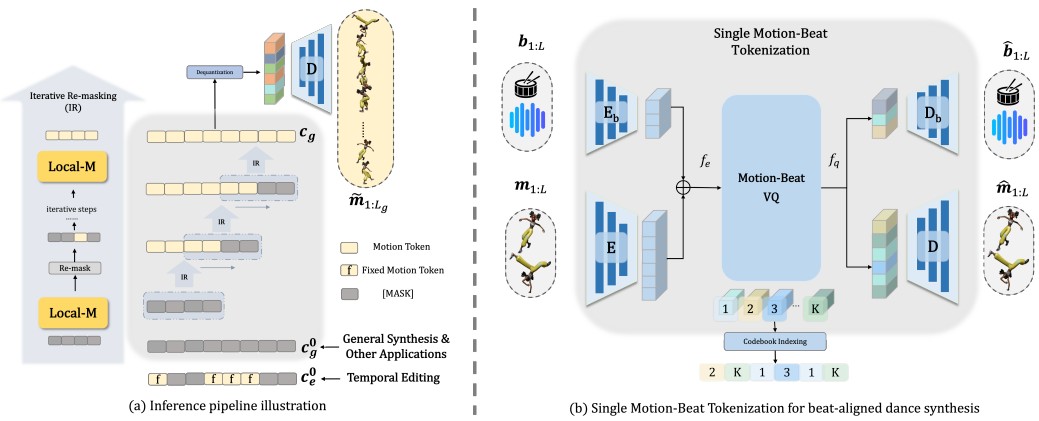

Figure 8: (a) Inference pipeline illustration. (b) Single Motion-Beat Tokenization for beat-aligned dance synthesis.

## A.2 IMPLEMENTATION DETAILS

The proposed method and architecture have been majorly illustrated in Section 3 of our paper. The involved parameter setting of the proposed architecture of MotionDreamer is presented in Table 4. During training, we crop the single reference motion sequence into overlapping motion patches of length $T_p$ and $s_p$. At the single motion tokenization phase, encoder $E$, codebook $\mathcal{C}$ and decoder $D$ are trained with learning rate $lr_1$. Local-M transformer is trained with learning rate $lr_2$. We select the parameter settings for training in Table 5 based on the best empirical results. All the reference motions are trained and evaluated on a single RTX2080Ti GPU.

Table 4: Parameter settings for architectures.

| Param. | Definition | Value |
|---|---|---|
| $h$ | down-sampling factor in tokenization | 8 |
| $d$ | latent embedding $f_e$, $f_q$ dimension | 4096 |
| $d_k$ | transformer latent embedding dimension | 384 |
| $N_E$ | number of layers in $E$ | 3 |
| $N_D$ | number of layers in $D$ | 3 |
| $N_M$ | number of layers in Local-M | 3 |
| $K$ | number of code entries in $\mathcal{C}$ | 48 |
| $W$ | local window size of SlidAttn layer | 5 (tokens) |
| $S$ | window stride of SlidAttn layer | 4 (tokens) |

Table 5: Parameter settings for training and inference.

| Param. | Definition | Value |
|---|---|---|
| $T_p$ | length of a motion patch for training VQ, Local-M | 96, 128 (frames) |
| $s_p$ | stride of motion patches for training VQ, Local-M | 24, 32 (frames) |
| $\beta_q$ | weighting factor of $\mathcal{L}_q$ | 0.1 |
| $\beta_k$ | weighting factor of $\mathcal{L}_{\text{token}}$ | 1e-3 |
| $\lambda_{\text{rec}}$ | weighting factor of $\mathcal{L}_{\text{rec}}$ during training Local-M | 0.2 |
| $lr_1$ | learning rate for training tokenization | 2e-4 |
| $lr_2$ | learning rate for training Local-M | 2e-4 |

## A.3 INFERENCE PROCESS ELABORATION

At the inference stage, shown in Figure 8, we synthesize the novel motion $\tilde{m}_{1:L_g}$ of arbitrary length $L_g$ in a localized auto-regressive strategy based on sliding windows. We first initialized a blank template token sequence $c_g^0$ of length $L_g/h$ with all the tokens masked. A synthesis window of size $W_g$ is preset, and slides along the template token sequence with stride $S_g$ for auto-regressive synthesis. At each window-step, the learned Local-M transformer progressively unmasks all the motion tokens within the synthesis window through iterative re-masking (IR), where tokens generated with low confidence are re-masked for updated generation in the next iteration. The stride of the sliding synthesis window ensures overlapping tokens from the previous window-step. These overlapping tokens remain un-masked as condition tokens for generating other tokens in the new window-step. After all the window-steps, the completely generated token sequence $c_g$ is de-quantized and decoded to motion space, resulting in novel motion $\tilde{m}_{1:L_g}$. Our best empirical selection for parameters is $W_g = T_p$, $S_g = 3T_p/4$.

## A.4 APPLICATION IMPLEMENTATION ELABORATION

**Crowd Animation**    The crowd animation shares exactly the same architecture, training and inference pipelines as what we describe in the major contents. What we showcase here in this application is the way to use MotionDreamer as an efficient tool for generating similar yet diverse motions for a crowd of characters with the same skeleton structures. More demos are shown in our project page link in A.1.

**Temporal Editing**    As presented in Figure 8(a), the template token sequence for temporal editing at the inference stage, noted as $c_e^0$, is constructed differently compared with either general synthesis elaborated in Appendix A.3 or other applications discussed in this section. By providing the sub-parts to generate with corresponding timestamps, the template sequence is constructed as the combination of encoded motion tokens for the fixed sub-parts and the [MASK] tokens for the sub-parts to be generated. The motion generated from $c_e^0$ retains exactly the same motion for the fixed sub-parts, while presenting diverse motions based on learned internal patterns for the others.

**Beat-aligned Dance Synthesis**    The beat-aligned dance synthesis involves incorporating auxiliary beat features into the latent motion representation, ensuring that the synthesized motion aligns with the rhythm while maintaining natural transitions and variations. To approach this, the single motion-beat tokenization is proposed where we establish the lightweight encoder($E_b$)-decoder ($D_b$) network based on only $N_E$ $(= N_D)$ 1D-convolutional layers for embedding the beat features, as shown in Figure 8(b). $E_b$ maps beat features, extracted from the paired music the using librosa (McFee et al., 2015), to the discrete latent space shared with the motion features; $D_b$ decodes the quantized beat embeddings back to the original space for beat features. This lightweight architecture facilitates tempo alignment based on single-instance while minimizing computational overhead.

## A.5 EVALUATION METRICS

The computations of our applied evaluation metrics follow Li et al. (2022a) and Raab et al. (2024).
**Coverage**    The coverage of the reference motion $m$ is measured based on all possible temporal windows given length $L_c$, which is written as $\mathcal{W}_{L_c}$. Given a generated motion $\tilde{m}$, a temporal window $m_w \in \mathcal{W}_L$ would be labeled as covered if the distance measure for the nearest neighbor in $\tilde{m}$ is smaller than a threshold $\epsilon$. The distance metric is chosen as Frobenius norm on the local joint rotation matrices, and the window size $L_c = 30$, defined as:

$$\text{Cov}(m, \tilde{m}) = \frac{1}{|\mathcal{W}_{L_c}|} \sum_{m_w \in \mathcal{W}_{L_c}} \mathbf{1}[\text{NN}(m_w, \tilde{m}) < \epsilon]. \tag{9}$$

**Global Diversity**    Li et al. (2022a) proposes a distance measure for patched nearest neighbors (PNN) to measure the global diversity. The generated motion is segmented with length no shorter than a threshold $T_m$ for each sub-part, and the measure is oriented for finding an optimized segmentation for the motion with minimum averaged per-frame nearest neighbor cost. For more details, we refer the readers to Li et al. (2022a)

**Local Diversity**    The local diversity metric is based on the local frame diversity by comparing every local window of the generated motion to its nearest neighbor in the reference motion:

$$D_{\text{Local}} = \frac{1}{|\mathcal{W}_{L_d}|} \sum_{\tilde{m}_w \in \mathcal{W}_{L_d}} \text{NN}(\tilde{m}_w, m). \tag{10}$$

## A.6 MORE ABLATION STUDIES

Table 6 shows the ablation study for our settings for the architecture and the key parameters.

**Local Attention Mechanism**    We want to highlight the second part of Table 6, where we compare our proposed SlidAttn with other two alternatives for local attention layer, Local SASA and QnA. Compared to Local SASA, which is the most standard local attention layer which processes non-overlapping windows without learnable queries or positional encodings, SlidAttn significantly improves coverage and diversity, demonstrating the importance of overlapping windows and enriched attention computation. QnA applied in SinMDM (Raab et al., 2024) is a convolutional local

Table 6: Ablation study for parameter settings. **Bold** text marks the best result.

| Method | Coverage (%) ↑ | Global Div. ↑ | Local Div. ↑ | Inter Div. ↑ | Intra Div. ↓ | Harmonic Mean ↑ |
|---|---|---|---|---|---|---|
| $K = 32$ | 84.23 | 1.08 | 1.02 | 0.16 | 0.30 | 0.24 |
| $K = 48$ | **93.47** | **1.33** | **1.17** | **0.25** | **0.28** | **0.43** |
| $K = 64$ | 73.36 | 1.20 | 1.09 | 0.14 | 0.34 | 0.22 |
| Local SASA | 74.23 | 1.13 | 1.10 | 0.15 | 0.31 | 0.23 |
| QnA (Raab et al., 2024; Arar et al., 2022) | 85.02 | **1.37** | **1.20** | **0.26** | 0.32 | 0.31 |
| SlidAttn | **93.47** | 1.33 | 1.17 | 0.25 | **0.28** | **0.43** |
| SlidAttn w/o. learnable queries | 71.42 | **1.48** | **1.37** | 0.20 | 0.36 | 0.23 |
| SlidAttn w/o. $\mathbf{r}$ | 89.84 | 1.05 | 1.01 | 0.22 | 0.30 | 0.22 |
| SlidAttn | **93.47** | 1.33 | 1.17 | **0.25** | **0.28** | **0.43** |
| $W = 3$ | 82.39 | 1.24 | 1.12 | 0.20 | 0.29 | 0.25 |
| $W = 5$ | **93.47** | 1.33 | 1.17 | **0.25** | **0.28** | **0.43** |
| $W = 7$ | 88.71 | **1.36** | **1.19** | **0.25** | 0.36 | 0.35 |
| $S = 1$ | 81.79 | **1.41** | **1.20** | 0.23 | 0.29 | 0.28 |
| $S = 4$ | 93.47 | 1.33 | 1.17 | **0.25** | **0.28** | **0.43** |
| $S = 5$ | **94.89** | 1.03 | 0.98 | 0.24 | 0.38 | 0.38 |

Table 7: Impact of codebook regularization loss on other VQ methods for motion representation. **Bold** text marks the best result.

| Method | R Precision Top 1↑ | Perplexity↑ |
|---|---|---|
| MMM (Pinyoanuntapong et al., 2024) | 0.503 | 1642.194 |
| MMM (Pinyoanuntapong et al., 2024) w/ $\mathcal{L}_{\text{token}}$ | **0.572** | **1678.538** |
| MoMask (Guo et al., 2024) | 0.504 | 368.914 |
| MoMask (Guo et al., 2024) w/ $\mathcal{L}_{\text{token}}$ | **0.504** | **372.702** |

attention layer modified from Arar et al. (2022), which incorporates learnable queries and relative positional encoding. Compared to QnA, which relies on the convolutional layers for overlapping window operation, SlidAttn reaches better overall performance with a dramatic rise in coverage measures and competitive diversity results. This demonstrates that our proposed SlidAttn uniquely employs a more effective sliding window attention computation and aggregation paradigm for processing overlapping local windows under this framework.

**Sliding Window Parameters** The last part is worth highlighting as it presents the ablations for stride of the sliding windows in SlidAttn of Local-M (refer to Section 3), which demonstrates that having a relatively small overlap length ($W - S = 1$) in our case results in the best Harmonic Mean, which attains good diversity while preserving the internal patterns from reference motion.

A.7 DISCUSSION ON CODEBOOK REGULARIZATION

We also conduct additional experiments on the broader impact of our codebook regularization loss on general motion tokenization trained on large datasets. We choose MMM (Pinyoanuntapong et al., 2024) and MoMask (Guo et al., 2024) to experiment with, as they share similar framework but with different VQ architectures. MMM (Pinyoanuntapong et al., 2024) tokenizes the motions with a standard VQ layer with a large codebook, while MoMask (Guo et al., 2024) incorporates residual VQ layers (Borsos et al., 2023; Martinez et al., 2014; Zeghidour et al., 2021) with a stack of codebooks for fine-grained motion representation. We apply our codebook regularization loss $\mathcal{L}_{\text{token}}$ to the two methods for training the motion tokenization, comparing 1) *R Precision Top 1* for the reconstruction performance or the motion representation capacity of learned codebook(s); 2) *perplexity* for the codebook utilization.

As the results shown in Table 7, both R precision and perplexity have been significantly improved for MMM, indicating better and more effective use of the codebook to become more effective at representing motion data. For MoMask, the impact of $\mathcal{L}$token is also evident. While the R precision remains stable at 0.504, the perplexity improves from 368.914 to 372.702. This shows that the incorporation of $\mathcal{L}$token helps optimize the utilization of residual VQ layers without sacrificing the motion representation capacity. These results clearly demonstrate the effectiveness of our proposed $\mathcal{L}_{\text{token}}$in enhancing the quality and utilization of codebooks across different VQ architectures, making it a valuable addition to motion representation frameworks.

The improvements are particularly significant for methods like MMM, where the standard VQ layers benefit the most from the regularization. For more advanced residual VQ architecture adapted

by MoMask, the improvements in codebook utilization are moderate, with no significant gains in motion representation capacity. This may be due to the nature of residual VQ layers, which obtain higher motion representation capacity by utilizing multiple codebooks to quantize the information loss from previous layers, rather than directly representing motion data. Consequently, higher codebook utilization alone does not significantly enhance motion representation capacity in this context.

There are also other alternatives regarding the codebook regularization for VQ-based representation, which can be further explored in future work. For example, entropy regularization, which encourages entropy maximization of the token distribution, can promote codebook utilization, as applied in Volkov (2022); Xiao et al. (2024). Stochastic sampling methods can also be incorporated such as Zhang et al. (2023b) by modifying sampling strategies to implicitly regularize the token utilization distribution across large-scale datasets.

## A.8    LIMITATIONS

While MotionDreamer demonstrates strong performance in generating diverse and natural motion sequences from a single reference, several limitations remain. First, the framework's performance is highly dependent on the quality of the reference motion. Second, due to the inherent nature of single instance based motion synthesis, the framework exhibits limited capacity to generalize across a broader range of motion editing and conditional synthesis tasks, given the nature of single instance based synthesis. Future work will focus on improving the model's extrapolation capabilities and enhancing its generalization to both few-shot and large-scale datasets. In particular, the development of a more flexible framework capable of distilling knowledge from diverse motion priors would allow the model to accommodate a wider array of motion styles and patterns. Moreover, integrating a more robust attention mechanism may enable the model to capture long-range dependencies and global patterns more effectively, thus extending its applicability to a broader set of motion editing and synthesis tasks.

