# OpenReview forum: "MotionDreamer: One-to-Many Motion Synthesis with Localized Generative Masked Transformer"
_ICLR.cc/2025/Conference — ICLR 2025 Poster_

### Official Review · Reviewer_Fzyk · 2024-11-02

**Soundness:** 3
**Presentation:** 4
**Contribution:** 2
**Rating:** 5
**Confidence:** 4

**Summary:**

This paper proposes a generative masked model (MotionDreamer) for motion synthesis domain based on the motivation "large datasets are not always available". MotionDreamer designs a local masked modeling paradigm to learn motion internal patterns from a given motion with a key sliding window local attention module. Experiments demonstrate various motion tasks, including crowd motion synthesis, temporal editing and beat-aligned dance synthesis.

**Strengths:**

- The paper is well-written and is easy to follow.

- Qualitatively results look impressive. Ablation studies prove the effectiveness of the designs.

- The authors introduce various applications of motion synthesis and achieve great performance.

**Weaknesses:**

- Generative masked modeling for motion synthesis has been investigated in many recent works, such as MMM, Momask, MotionGPT, as described in paper. On the other hand, single motion synthesis is also not new, such as SinMDM. It seems that MotionDreamer integrates the techniques of generative masked modeling into the task of single motion synthesis, including motion tokenization and codebook distribution regularization.

- Capturing local dependencies of motion features is quite important for single motion synthesis. SinMDM introduces QnA layers that allow local attention with a temporally narrow receptive field. And MotionDreamer incorporates sliding window local attention to achieve this. Besides, another key design of codebook distribution regularization has been explored in single image synthesis domain. Therefore, I think the technical contribution of the paper is somewhat limited.

-  In Ablation Studies (Sec. 4.4), it's better to use QnA layers in SinMDM to replace sliding window local attention and show the comparison of the performance. Due to the same task, the paper should demonstrate its unique effectiveness.

**Questions:**

See Weakness.

---

> ### Author Response · Authors · 2024-11-22
> **Response to Reviewer Fzyk (W1)**
>
> Thank you for the thoughtful comments and suggestions on our paper. Below are our answers to the raised questions.
>
> ---
>
> **W1**: *“Generative masked modeling for motion synthesis has been investigated in many recent works, such as MMM, Momask, MotionGPT, as described in paper. On the other hand, single motion synthesis is also not new, such as SinMDM. It seems that MotionDreamer integrates the techniques of generative masked modeling into the task of single motion synthesis, including motion tokenization and codebook distribution regularization.”*
>
> **A1**: We appreciate the opportunity to clarifying our motivation and contribution. MotionDreamer is not a simple integration of techniques to adapting generative masked modelling. Either the choice of this framework or the set of non-trivial technical adaptations introduced is fundamentally motivated by our observations to the major issues in the single motion synthesis task rather than random picking. Below is the elaboration:
>
> The underlying core of single motion synthesis is to **learn a nice representation of internal motion patterns for a single motion sequence of arbitrary length and structure**, which is worth exploring in that it can also provide insights for related motion synthesis task, e.g. motion composition [1], long-term motion synthesis [2, 3],  where representation and synthesis of natural local motion patterns is also important for seamless transitions. Such a representation should be able to faithfully reproduce each patterns while coherently diversify the seamless combinations of them at the same time.  Existing methods like SinMDM and GenMM have yet to fully address this balance between faithfulness and diversity (Line 034-044), which has been shown in our quantitative (Table 1.) and qualitative (Figure 5 and Appendix A.1) comparison. This motivates us to develop a more robust representation that achieves stronger generalization than the straight-through motion patch-based GenMM for greater diversity, while providing a more explicit modelling of internal patterns than the implicit representations used in SinMDM, enabling more faithful preservation of reference motion patterns.
>
> In this context, generative masked modelling framework naturally presents as a good fit because of its inherent ability to learn discrete latent representations and model explicit categorical distributions. While MotionDreamer is built upon the generative masked modelling framework, we introduces critical and non-trivial adaptations that are essential for its success in single motion synthesis particularly:
>
> - **Codebook Regularization for Single Motion Tokenizaiton**: We observed that single motion tokenization is more prone to codebook under-utilization (Line 204 - 207 and Figure 5.), where common strategies applied in MMM, MoMask, MotionGPT are insufficient for learning a meaningful codebook for single motion sequence. MotionDreamer introduces a KL-divergence-based codebook regularization loss (Eq. 4), which significantly improves code entry utilization and fidelity of learned patterns (Table 2., Figure 5 and Appendix A.1).
> - **Sliding Window Local Attention (SlidAttn)**: Standard transformer blocks used in common generative masked modelling, as seen in MMM and MotionGPT, suffer from overfitting to global patterns when adapted to single sequences. MotionDreamer introduces SlidAttn (Sec. 3.2.1), which captures the local dependencies and models local transitions effectively. The overlapping windows and overlap attention fusion (AttnFuse) ensure smooth pattern transitions without losing coherence (Table 3, Figure 6 and Appendix A.1).
> - **Differentiable Dequantization loss**: To bridge the discrete latent space and continuous motion space, MotionDreamer incorporates a sparsemax-based differentiable dequantization loss (Eq. 7), which further enhance the temporal coherence and coverage of internal patterns (Table 3).
>
> [1] Barquero, G., Escalera, S., & Palmero, C. (2024). Seamless Human Motion Composition with Blended Positional Encodings. *Proceedings of the IEEE/CVF Conference on Computer Vision and Pattern Recognition (CVPR)*.
>
> [2] Wang, H., Zhang, X., Yuan, Y., Liu, Z., & Yu, C. (2021). Synthesizing Long-Term 3D Human Motion and Interaction in 3D Scenes. *Proceedings of the IEEE/CVF Conference on Computer Vision and Pattern Recognition (CVPR)*, 6195-6204.
>
> [3] Zhang, Z., Liu, R., Aberman, K., & Hanocka, R. (2024). TEDi: Temporally-Entangled Diffusion for Long-Term Motion Synthesis. *ACM Transactions on Graphics (TOG)*, 43(4), Article 123.

---

> ### Author Response · Authors · 2024-11-22
> **Response to Reviewer Fzyk (W2, W3)**
>
> **W2**: “ *In Ablation Studies (Sec. 4.4), it's better to use QnA layers in SinMDM to replace sliding window local attention and show the comparison of the performance. Due to the same task, the paper should demonstrate its unique effectiveness.* ”
>
> **A2**: Thank you for the suggestions. Following your comment, we have included the comparison results in our updated versions shown in the second subpart in Table 6 and further analyzed in Appendix A.6, and quantitative results are also shown as below.  Compared to QnA, which relies on the convolutional layers for overlapping window operation, SlidAttn reaches better overall performance with a dramatic rise in coverage measures and competitive diversity results.  It is worth noting that QnA uses a simplified linear attention without softmax, which prioritizes computational efficiency. However, the lack of non-linearity may limit its ability to model correlations effectively, resulting in lower coverage, as some key relationships in motion patterns cannot be fully captured.
>
> | **Method**           | **Coverage (%) ↑** | **Global Div. ↑** | **Local Div. ↑** | **Inter Div. ↑** | **Intra Div. ↓** | **Harmonic Means ↑** |
> |-----------------------|---------------------|--------------------|-------------------|-------------------|------------------|-----------------------|
> | Local SASA            | 74.23              | 1.13               | 1.10             | 0.15             | 0.31            | 0.23                 |
> | **QnA** *(SinMDM)*    | 85.02              | **1.37**           | **1.20**         | **0.26**         | 0.32            | 0.31                 |
> | **SlidAttn**          | **93.47**          | 1.33               | 1.17             | 0.25             | **0.28**        | **0.43**             |
>
> The updated visualization results present a qualitative comparison. As shown in the video, compared to SlidAttn, generations from QnA-based ablation tend to repetitively produce a certain few internal patterns, and lose track of the motion details.
>
> These results demonstrates that our proposed SlidAttn uniquely employs a more effective sliding window attention computation and aggregation paradigm under this framework for single motion synthesis.
>
> ---
>
> **W3**: “ *Capturing local dependencies of motion features is quite important for single motion synthesis. SinMDM introduces QnA layers that allow local attention with a temporally narrow receptive field. And MotionDreamer incorporates sliding window local attention to achieve this. Besides, another key design of codebook distribution regularization has been explored in single image synthesis domain. Therefore, I think the technical contribution of the paper is somewhat limited.*”
>
> **A3**: We would like to highlight our contribution regarding our proposed 1) sliding window local attention 2) codebook regularization for single motion tokenization.
>
> ***Contribution of Sliding Window Local Attention (SlidAttn)***
>
> While SinMDM’s QnA layers provide an effective approach using convolutional local attention with a narrow receptive field, our SlidAttn introduces significant enhancements for addressing this challenge. As shown in Table 6 and analyzed in Appendix A.6, SlidAttn improves the overall performance compared to QnA, especially in effectively capturing the internal patterns as shown by the rise in the coverage metrics. And the qualitative result also demonstrates that SlidAttn can lead to more perceptually natural and diverse motion generations aside from faithfully preserving the internal patterns. These improvements stem from SlidAttn's  **unique sliding window computation and aggregation paradigm**, which explicitly operate overlapping local windows, ensuring smoother transitions and better correlation across internal motion patterns.
>
> ***Contribution of Codebook Regularization for Single Motion Tokenization***
>
> We would like to point out that the work (Zhang et al., 2023b) we refer to in Section 3.1.1  addresses codebook regularization on large-scale image datasets, which is not designed for the single image synthesis task. In contrast, as elaborated in Section 3.1.1, MotionDreamer’s codebook regularization tackles a fundamentally different challenge: ensuring effective tokenization for a **single instance (motion in our case)**. Unlike general synthesis tasks based on large scale datasets, the token diversity is naturally constrained by the limited input for single-instance tasks. The effectiveness of the codebook regularization proposed in MotionDreamer (presented in Table 2. and Figure 5.) brought the insights that tokenization for single instance impressively benefits from explicit distribution regularization aside from implicit regularization such as EMA and codebook reset.
>
> ---
>
> *We sincerely hope that our answers have addressed your concerns, and we are happy to take any further questions.*

---

> ### Author Response · Authors · 2024-11-25
> **Additional Response to Reviewer Fzyk**
>
> Dear Reviewer Fzyk,
>
> We sincerely thank you again for your valuable comments and suggestions on our work, especially for the additional evaluation on our proposed SlidAttn layer compared with other previous established local attention layers on the single motion synthesis task. We hope that our revisions and updates following your suggestions have improved the clarity and soundness of our paper. *This is a kind reminder that the author-reviewer discussion will end in two days.* We are happy to take any further questions and comments for discussion. Please feel free to leave comments.
>
> Best,
>
> Paper 9141 Authors

---

> > ### Comment · Reviewer_Fzyk · 2024-11-26
> > **Response to Authors**
> >
> > Thanks for the reply. The authors have addressed some of my concerns. However, I still think that the integration of techniques in image domain into motion synthesis is somewhat straightforward and lacks originality that suits single motion synthesis (as is also pointed out by Reviwer oiwR).

---

> > > ### Author Response · Authors · 2024-12-02
> > >
> > > Dear reviewer Fzyk,
> > >
> > > We sincerely thank you for providing valuable feedback for our work, and we have made our responses correspondingly in our last reply. We hope that the responses can help address your concerns. ***The author-reviewer discussion phase will end in less than 24 hours.*** Please feel free to let us know if you have any further comments. If our responses and updates have properly addressed your concerns, we would really appreciate it if you could kindly re-evaluate our work. Thank you again for your time and efforts!
> > >
> > > Best,
> > >
> > > Paper 9141 Authors

---

> ### Author Response · Authors · 2024-11-28
> **Thank you for the follow up feedback**
>
> Dear Reviewer Fzyk,
>
> *Thank you for your follow up feedback on our previous response. We hope that the following would further address your concerns.*
>
> ---
>
> We would like to respectfully point out that MotionDreamer is not a simple integration of techniques in image domain into motion synthesis, neither from the overall picture of our framework nor the fine-grained technical contributions.
>
> - **The generative masked modelling for the single instance based synthesis is not only novel for motions, but also never seen in the similar tasks for images** *(mentioned in our response to Reviewer oiwR)*. As explained in our introduction, the adaptation of generative masked modelling naturally comes from the inadequate representation of previous methods *(line 034-044)*, and the strength of explicit distribution modelling *(line 045-050)*.  **MotionDreamer learns the explicit categorical distribution via generative masked modelling on a single instance for the first time (a single motion sequence in our case), and demonstrates a much more comprehensive performance on the single motion synthesis task** with great diversity while preserving faithfulness to the in-distribution motion patterns and motion naturalness at the same time *(Table 1, Figure 3, Appendix A.1)*.
> - Unlike continuous latent spaces, which can introduce out-of-distribution embeddings during inference even when constrained during training, the explicit discrete representation in MotionDreamer inherently ensures in-distribution generations. This fundamentally guarantees the faithfulness to the internal patterns of the single reference instance, and the naturalness of generated instances, which are highlighted aspects in the single instance based synthesis task. These insights about the latent representation can potentially benefit a broader range of tasks in motion synthesis *(mentioned in our previous response to W1)*, and similar tasks in image synthesis too.
> - From the fine-grained technical perspective, the key contributions are well motivated and designed from task-specific observations and assumptions, rather than being a simple “mishmash” of techniques borrowed from image-domain methods.
>     - **[ Codebook Regularization for Single Motion Tokenizaiton ]**      While the proposed KL-divergence-based regularization draws inspiration from image-domain techniques, it directly addresses critical challenges unique to single motion synthesis. The single motion sequences itself often result in imbalanced distributions of internal patterns from temporal perspective, where repetitive patterns can easily dominate, leaving rare but critical motion segments insufficiently captured. **This inherent imbalanced temporal distribution of internal patterns also exacerbates the risk of codebook collapse aside from the limited data scale** *(elaborated in Section 3.1.1, line 200-201; further explained in our previous response to W1 & W3)*, which in turn limits the ability to represent the internal motion patterns of the single reference motion faithfully.  **The proposed KL-divergence-based regularization mitigates this by enforcing balanced token utilization, ensuring that scarce yet important patterns are adequately captured.** Results in *Figure 3* can further back up this statement in addition to *Table 2 and Figure 5*, where MotionDreamer can well preserve highly dynamic yet typical internal patterns with high fidelity,  while previous methods often fall short in.
>     - **[ Sliding Window Local Attention (SlidAttn) ]**      As we both agree for the single motion synthesis, the task inherently relies on capturing localized temporal dependencies to maintain temporal coherence across internal motion patterns. SlidAttn is uniquely designed for this purpose,  **introducing a novel local attention paradigm of an explicit sliding temporal windows strategy along with a specific attention fusion mechanism**. *Table 1, Table 6 and Appendix A.6* have already shown the effectiveness of the design to the task.
>     - **[ Differentiable Dequantization loss ]**      Our sparsemax-based de-quantization loss is novelly designed to bridge the discrete latent space and continuous motion space during training, so as to enhance the temporal coherence and coverage of internal patterns *(effectiveness also shown in Table 3)*. **This is also unseen in any previous works no matter in motion nor image domain.**
>
> ---
>
> ***We sincerely appreciate the opportunity for further in-depth discussion on the contribution of our work. Please feel free to let us know if our response helps address your concerns or not. We're happy to open further discussion if you have any further questions or concerns.***
>
> Best,
>
> Paper 9141 Authors

---

### Official Review · Reviewer_oiwR · 2024-11-04

**Soundness:** 3
**Presentation:** 3
**Contribution:** 2
**Rating:** 3
**Confidence:** 4

**Summary:**

This paper proposes a localized masked modeling paradigm designed to learn motion internal patterns from a given motion with arbitrary topology and duration. It introduces the codebook distribution regularization and sliding window local attention (SwinAttn) mechanism to avoid codebook collapse and overfitting.

**Strengths:**

The codebook distribution regularization effectively prevents codebook collapse. The sliding window local attention mechanism in the Local-M transformer captures local dependencies and ensures smooth transitions across overlapping windows, enhancing the fidelity and diversity of generated motions.

**Weaknesses:**

This paper applies some recent advancements from the image domain to motion synthesis, which is a commendable attempt. However, the contribution feels somewhat limited. As a work focused on enhancing the VQVAE structure for motion synthesis, the authors might benefit from incorporating insights and experiments from NCP, rather than simply adapting the network architecture.
# Added after AC's comment
I'm grateful for the feedback and suggestions. To clarify, the aspects of the contribution that feel limited include the novelty in adapting image domain advancements without significant modification or integration specific to motion synthesis. Additionally, while some metrics have been provided, they may not fully reflect the method's advantages. I believe visualizations of the latent space could be helpful in demonstrating its effectiveness. Regarding NCP, I apologize for the oversight—it refers to Neural Categorical Priors for Physics-Based Character Control. Thank you for the guidance.

**Questions:**

In the final section of the paper, the authors mention several other applications. Could the authors elaborate on how the network architecture performs and what advantages it offers for these tasks?

---

> ### Author Response · Authors · 2024-11-22
> **Response to Reviewer oiwR (W1)**
>
> Thank you for your time and efforts reviewing our work. Here’s our answers to your questions:
>
> ---
>
> **W1**: *“This paper applies some recent advancements from the image domain to motion synthesis, which is a commendable attempt. However, the contribution feels somewhat limited. As a work focused on enhancing the VQVAE structure for motion synthesis, the authors might benefit from incorporating insights and experiments from NCP, rather than simply adapting the network architecture.”*
>
> **A1**: MotionDreamer indeed utilizes VQVAE as the backbone of single motion tokenization (Section 3.1), which is the first part of the whole paradigm.  Yet the main focus is not about enhancing VQVAE structure. We would like to remind the reviewer of our major contribution, which is also elaborated and summarized in Section 1:
>
> MotionDreamer is the first work leveraging the generative masked modelling framework for single instance based synthesis task, where we specifically deal with single motion instance of arbitrary skeleton and length. We train the model to learn explicit categorical distribution of the internal patterns of **the single reference motion**, from which natural and diverse motions preserving the internal patterns can be generated. Our method first tokenizes the internal motion patterns into discrete latent space, and models their local dependencies through **a novel masked transformer architecture**, ensuring explicit pattern representation and coherent transitions. It introduces non-trivial technical adaptations tailored for single-instance-based motion synthesis:
>
> - **Codebook Regularization for Single Motion Tokenizaiton**: We observed that single motion tokenization is more prone to codebook under-utilization (Line 204-207 and Figure 5.), where common strategies are insufficient for learning a meaningful codebook for single motion sequence. MotionDreamer introduces a KL-divergence-based codebook regularization loss (Eq. 4), which significantly improves code entry utilization and fidelity of learned patterns (Table 2., Figure 5 and Appendix A.1).
> - **Sliding Window Local Attention (SlidAttn)**: Standard transformer blocks used in common generative masked modelling suffer from overfitting to global patterns when adapted to single sequences. MotionDreamer introduces SlidAttn (Sec. 3.2.1), which captures the local dependencies and models local transitions effectively. The overlapping windows and overlap attention fusion (AttnFuse) ensure smooth pattern transitions without losing coherence (Table 3, Figure 6 and Appendix A.1).
> - **Differentiable Dequantization loss**: To bridge the discrete latent space and continuous motion space, MotionDreamer incorporates a sparsemax-based differentiable dequantization loss (Eq. 7), which further enhance the temporal coherence and coverage of internal patterns (Table 3).
>
> Regarding Neural Categorical Priors (NCP), it also utilizes VQVAE as a bottleneck for learning a categorical prior to facilitate the imitation policy for physics-based character control, which is trained on large motion datasets ( motion clips of around 30 minutes in total ) for each of the 2 pre-defined humanoid skeletons. It differs fundamentally from MotionDreamer’s task which focuses on motion synthesis from training on **a single reference motion** with **arbitrary skeletons**. Thus, the paradigm NCP provides hardly address any specific challenges we’re facing, such as localized pattern representation and diversity in single-instance settings.
>
> As for the technical details, the novel insights brought by NCP are more focused on integrating the discrete categorical prior modelled by VQ-VAE into the imitation policy, and training techniques for RL-based methods, which is not compatible with the framework suitable for the single-instance setting. Therefore, it’s hard to incorporate anything from NCP to our current work.

---

> ### Author Response · Authors · 2024-11-22
> **Response to Reviewer oiwR (Q1)**
>
> **Q1**: *“In the final section of the paper, the authors mention several other applications. Could the authors elaborate on how the network architecture performs and what advantages it offers for these tasks?”*
>
> **A1**: To address the concern, we’ve added a new section ( Appendix A.4 ) elaborating the implementation details of the applications in the updated version.
>
> - **Crowd Animation**: MotionDreamer is used for crowd animation by efficient generating diverse yet coherent motions for multiple characters sharing the same skeleton structure, when given a single reference motion. The architecture, training, and inference pipelines remain unchanged from the main method, showcasing its efficiency and flexibility for applying to real animation production. Additional demos are provided on the project page.
> - **Temporal Editing**: Temporal editing generates motion sequences by masking out only the selected sub-parts to be generated while fixing the other sub-parts with encoded motion tokens aside from . This approach ensures the fixed sub-parts remain unchanged while the generated segments retain diversity, guided by learned internal patterns.
> - **Beat-Aligned Dance Synthesis**: Beat-aligned dance synthesis incorporates auxiliary beat features into the latent motion space, aligning generated motions with the music rhythm. A lightweight encoder-decoder network embeds beat features into the shared discrete latent space, facilitating tempo alignment while minimizing computational overhead.
>
> ---
>
> *We sincerely hope that our answers have addressed your concerns, and we’re happy to take any further questions.*

---

> > ### Comment · Reviewer_oiwR · 2024-11-26
> >
> > Dear authors, thank you for your detailed response and clarification. My previous mention of NCP was intended to suggest adding more experimental analyses, such as latent space visualizations, to further enhance the insights and completeness of your work, rather than to propose a direct comparison with NCP. The additional experiments and your detailed response are meaningful and demonstrate a deep understanding of the single-instance-based synthesis task. However, I still feel that the overall contribution could be further strengthened.

---

> ### Author Response · Authors · 2024-11-25
> **Additional Response to Reviewer oiwR**
>
> Dear Reviewer oiwR,
>
> We sincerely thank you for your review on our work. *This is a kind reminder that the author-reviewer discussion period will be over in two days.* Please don't hesitate to let us know if you have any further questions or comments.
>
> Best,
>
> Paper 9141 Authors

---

> ### Author Response · Authors · 2024-12-01
> **Thank you for follow up comments**
>
> Dear Reviewer oiwR,
>
> ***Thank you for your follow up feedback on our previous response. We appreciate the opportunity to provide  additional analysis to back up our contributions.***
>
> ---
>
> Following your suggestion, we visualize the discrete latent space learned by MotionDreamer in ([Figure a1.](https://drive.google.com/file/d/1vpkuBOxqhXX2d9GSl_rI_RvOO0ZLiO1a/view?usp=sharing)), and the continuous latent space learned by SinMDM as a comparison in ([Figure a2.](https://drive.google.com/file/d/1oMxoeiVgtpCoosB0cHc_b4FJTzfyYqRu/view?usp=sharing)).
>
> - For continuous latent space learned for single-instance-based task, e.g. SinMDM shown in Figure a2, the local components related to temporal cues are implicitly modelled which obscure the learned internal patterns. Moreover, for a single instance, the embeddings seem to over spread across the space with potential out-of-distribution points, which may lead to lack of faithful representation of reference internal patterns.
> - In contrast, with the explicit categorical distribution learned via motion tokens, MotionDreamer enables **visualization of explicit internal pattern modes** by specifying a granularity with a proper size of temporal local windows, as shown by the dots in Figure a1. The proximity measures of generated modes (blue) to their closest reference modes (red) further demonstrate high coverage of the internal motion patterns of the reference motion, with very few outliers observed. Additionally, the span of each reference mode with its proximate generated ones indicates the diversity of generations. We also refer you to *Figure 2(b)* in the main paper, where we qualitatively visualize the mechanism and performance of explicit distribution modeling for the internal patterns of a single motion. This serves as complementary evidence of the effectiveness of our learned latent space, reinforcing its robustness and interpretability in representing internal motion patterns.
>
> **The success of learning the explicit modelling of internal patterns of a single motion not only facilitates a more precise representation of the internal motion patterns but also makes it easier to disentangle, interpret, and manipulate for potential uses.** This capability further highlights the strengths of MotionDreamer in the single instance based motion synthesis on enabling interpretability and controllability for downstream tasks.
>
> With the updates, we would like to kindly reiterate the well-demonstrated novel contributions of MotionDreamer *(both overall picture and detailed techniques, also further elaborated in our additional response to reviewer Fzyk)*:
>
> - **Explicit Categorical Representation:** By leveraging generative masked modeling, MotionDreamer introduces explicit modeling of internal motion patterns for the single motion in a discrete space, achieving diversity without compromising faithfulness to reference motions, which also enables interpretable and controllable manipulation for downstream tasks. Figure a1  in this response and Figure 2(b) in the main paper demonstrate this.
> - **Localized Dependency Modeling:** Sliding Window Local Attention (SlidAttn) as a novel attention layer, ensures coherent transitions and captures localized dependencies better than standard attention mechanisms in image domains. *Table 1, Table 6 and Appendix A.6* have already shown the effectiveness of the design to the task.
> - **Codebook Regularization for Single Motion Tokenization:** The proposed codebook regularization well addresses token distribution biases unique to single motion sequences for the first time, balancing token utilization and enhancing representation capacity.
> - **Differentiable Dequantization loss**: Our sparsemax-based de-quantization loss is novelly designed to bridge the discrete latent space and continuous motion space during training, so as to enhance the temporal coherence and coverage of internal patterns *(effectiveness also shown in Table 3)*.
>
> We find the latent space analysis as an interesting experimental back up for our contributions, and thank you again for raising the comments about this! Due to the restrictions for submission, we are not able to update that into our paper for now. If the paper get accepted, we will include this part of analysis and discussion in our final version.
>
> ---
>
> ***Please feel free to let us know if you have further comments. If our responses and updates have properly addressed your concerns, we would really appreciate it if you could kindly re-evaluate our work. Thank you again for your valuable feedback and all the efforts.***
>
> Best,
>
> Paper 9141 Authors

---

> ### Author Response · Authors · 2024-12-02
>
> Dear reviewer oiwR,
>
> Thank you so much for providing follow up feedback for our work. We have made our responses in our last reply, and we hope that the responses can help address your concerns. ***The author-reviewer discussion phase will end in less than 24 hours.*** Please feel free to let us know if you have any further comments. If our responses and updates have properly addressed your concerns, we would really appreciate it if you could kindly re-evaluate our work. Thank you again for your time and efforts!
>
> Thanks,
>
> Paper 9141 Authors

---

### Official Review · Reviewer_L4V3 · 2024-11-04

**Soundness:** 3
**Presentation:** 2
**Contribution:** 3
**Rating:** 8
**Confidence:** 3

**Summary:**

The paper proposes Motiondreamer, a framework for one-to-many motion synthesis. The paper aims to learn the underlying motion pattern in a given sequence and leverage the motion pattern to generate natural and diverse motions.
The general idea of the paper is to first quantize a motion sequence into tokens, and then train a masked transformer to generate these motion tokens.
There are two core technical contributions,  one lies in regularising motion tokenization to encourage more coverage, and the other one lies in injecting localality into the makes transformer to encourage motion decomposition.

**Strengths:**

- The model is well-designed. Given the scarcity of motion data in this one-to-many setting, the inductive bias in the model design is particularly important. The overall framework follows a masked transformer architecture which is simple, but multiple practical techniques have been contributed to balance the motion faithfulness and the motion diversity, including
    - A KL term is added to the quantizer training besides the standard commitment loss to encourage coverage.
    - A sliding window attention that adds locality to the transformer to encourage diversity.
    - A differentiable dequantization loss that helps to compute loss on the motion space.

    All of them are well-motivated and make sense to me.

- The ablation study is very comprehensive. Although the above techniques, like KL regularisation, are quite standard and empirical, the paper provides detailed experiments to support their importance to the final results. I find those experiments convincing.
- The paper shows qualitative results on a webpage, which generally looks good to me. Compared to SinMDM, the proposed method reduces unnatural motions. Compared to SinMDM, the generated motion seems more diverse.

**Weaknesses:**

- Table 1 and Figure 3 are inconsistent in the comparison between GenMM with Ours. Table 1 shows that GenMM has much better coverage than Ours. However, in Figure 3, the failure mode of GenMM is missing a part of the reference motion.
- The beat-aligned dance synthesis application does not convince me. The generated dances have a clear mismatch with the tempo. Most importantly, the application (including implementation details) is not well documented in the paper so I am not fully sure what is happening there, for example, what is a “light-weight encoder-decoder”? To my understanding, the beat-aligned dance is like an extension of the “subpart generation” task while keeping predefined keyposes on the music beats. That is not sufficient to produce a beat-aligned dance in my opinion.
- Some of the techniques could be a little niche to be presented as part of the method. For example, AttnFuse could better fit into implementation details.
- Minor
    - L281: by progressively fill → by progressively filling.

----
The overall framework makes sense to me. The proposed techniques are simple, but they are well-motivated and are supported by solid ablation experiments. I am convinced that they contribute to the overall performance. My concerns are about the comparison between GenMM and the dance generation application part. The presentation can be improved as well. I am slightly inclined to accept.

**Questions:**

Please address the weaknesses above.

---

> ### Author Response · Authors · 2024-11-22
> **Response to Reviewer L4V3 (W1)**
>
> We sincerely appreciate the positive feedback on our well-motivated model design, comprehensive experiments and qualitative results, and thank you for kindly pointing out several issues in concerns. Our response to the comments can be found below.
>
> ---
>
> **W1**: *“Table 1 and Figure 3 are inconsistent in the comparison between GenMM with Ours. Table 1 shows that GenMM has much better coverage than Ours. However, in Figure 3, the failure mode of GenMM is missing a part of the reference motion.”*
>
> **A1**: Thank you for the comment regarding Table 1 and Figure 3. We would like to point out that Table 1 and Figure 3 are actually consistent and complementary results for comparing GenMM with our method. The quantitative coverage metrics in Table 1 and the qualitative presentation in Figure 3 reflect two non-contradictory aspects of how well the model preserves the reference internal patterns.
>
> - **Quantitative coverage metrics in Table 1**: As defined in Appendix A.5 (Eq. 9), the coverage metrics measures the proportion of temporal windows  $m_w$  in the reference motion  m  that are matched by the nearest neighbor in the generated motion  $\tilde{m}$  within a threshold  $\epsilon$ . It focuses on **statistically quantifying** whether the generated motion broadly covers the reference internal patterns. However, such metrics tend to overlook **the actual perceptual fidelity**, particularly for highly dynamic internal patterns that are brief in duration but visually significant or typical in the reference motion.
> - **Visual fidelity evaluation in Figure 3**: In this context, Figure 3 serves as a complementary perspective to evaluate visual fidelity. As shown in Figure 3 and supplementary video demos for diversity comparison (webpage link in Appendix A.1), the higher coverage measures for GenMM does not necessarily translate to high-fidelity synthesis of preserving the typical patterns of motions, especially those highly dynamic yet typical patterns such as pattern A. As shown, GenMM loses track of pattern A in their generations while MotionDreamer achieves better performance by preserving such patterns with fewer failure modes.
>
> We have revised the analysis in Section4.2-Qualitative Comparison by summarizing our answers above.

---

> ### Author Response · Authors · 2024-11-22
> **Response to Reviewer L4V3 (W2, W3, W4)**
>
> **W2**: *"The beat-aligned dance synthesis application does not convince me. The generated dances have a clear mismatch with the tempo. Most importantly, the application (including implementation details) is not well documented in the paper so I am not fully sure what is happening there, for example, what is a “light-weight encoder-decoder”? To my understanding, the beat-aligned dance is like an extension of the “subpart generation” task while keeping predefined keyposes on the music beats. That is not sufficient to produce a beat-aligned dance in my opinion.”*
>
> **A2**: We would like to first answer the last two question regarding the implementation details of this application to give a sufficient background of answering the first comment about the performance.
>
> - Implementation details for beat-aligned dance synthesis have been included in Appendix A.4 in the updated paper submission. This application involves incorporating auxiliary beat features into the latent motion representation, ensuring that the synthesized motion aligns with the rhythm while maintaining natural transitions and variations.  The “light-weight encoder-decoder” refers to the shallow encoder-decoder network based on very few 1D-convolutional layers used for embedding the beat features.   By efficiently embedding beat features, this lightweight architecture facilitates tempo alignment based on single-instance while minimizing computational overhead.
>
> - This approach is not a simple extension of the “subpart generation” task, as it embeds beat features directly into the discrete latent representation during training to ensure the alignment while maintaining natural motion transitions , rather than directly fixing predefined key poses at inference stage. As Figure 7(c) also shown, the two synthesized dance motions automatically diversify the key poses at each beat stress instead of simply fixing the corresponding ones from the reference dance.
>
> Achieving fine-grained beat alignment is a challenging goal, which typically involves training on large scale music-motion datasets [1, 2, 3]. With the constraint of single instance learning,  it is hard to achieve fine-grained, perfect beat-alignment. Instead, given a single reference motion that originally aligns with a given paired music, this application focuses on **generate tempo-correlated dance motions with learned internal patterns** while balancing **coherence** and **diversity**. As demonstrated in Figure 7(c) and the visualization result in (webpage link in Appendix A.1), MotionDreamer is able to preserve the internal dance patterns with awareness of the alignment to beat features, and introduce diversity in presenting the patterns. While some slight mismatches with the tempo may occur, the generated dance motions capture the overall temporal alignment with musical beats.
>
> ---
>
> **W3**: *"Some of the techniques could be a little niche to be presented as part of the method. For example, AttnFuse could better fit into implementation details.”*
>
> **A3**: Thank you for sharing your suggestion on the presentation of AttnFuse. We feel it may be better to include it in Section 3 due to following reasons:
>
> - Ensuring the completeness and coherence of the description of our approach.
> - Though it is not our major technical contribution, it plays an important role in our framework by ensuring smooth transitions. It contributes to the model’s performance as reflected in Section 4.4 and Figure 6, where it demonstrates improved diversity and fidelity compared to standard pooling methods. If the reviewer feels it necessary, we are open to condensing its description.
>
> ---
>
> **W4**: *"Minor typo: e.g. L281: by progressively fill → by progressively filling.”*
>
> **A4:** Thank your for the kind reminder of the existing typo. We have made the revisions in the updated version of our paper.
>
> ---
>
> [1] Li, R., Yang, S., Ross, D. A., & Kanazawa, A. (2021). AI Choreographer: Music Conditioned 3D Dance Generation with AIST++. *Proceedings of the IEEE/CVF International Conference on Computer Vision (ICCV)*, 13401-13412.
>
> [2] Tseng, J., Castellon, R., & Liu, C. K. (2023). EDGE: Editable Dance Generation from Music. *Proceedings of the IEEE/CVF Conference on Computer Vision and Pattern Recognition (CVPR)*, 448-458.
>
> [3] Bhattacharya, A., Paranjape, M., Bhattacharya, U., & Bera, A. (2024). DanceAnyWay: Synthesizing Beat-Guided 3D Dances with Randomized Temporal Contrastive Learning. *Proceedings of the AAAI Conference on Artificial Intelligence*, 38(2), 783-791.
>
> ---
>
> *We sincerely hope that our answers have addressed your concerns, and we are happy to take any further questions.*

---

> ### Author Response · Authors · 2024-11-25
> **Additional Response to Reviewer L4V3**
>
> Dear Reviewer L4V3,
>
> Thank you again for your thorough review and suggestions. We believe that your comments have helped further strengthen the empirical evaluation and the soundness of our work. *The author-reviewer discussion will be closed in two days.* Please feel free to let us know if you have any further comments, and we would be more than happy to discuss on them.
>
> Best,
>
> Paper 9141 Authors

---

> > ### Comment · Reviewer_L4V3 · 2024-11-28
> >
> > Thanks for the response. The authors have fully resolved my concerns about the result inconsistency and the beat-aligned dance. I have also read the comments from other reviewers. I agree with Reviewer Fzyk that the technical novelty is not the strength of this paper. However, the framework looks well thought out and the performance can be backed up with extensive experiment results. Therefore I move my rating from borderline accept to accept.

---

> > > ### Author Response · Authors · 2024-11-28
> > >
> > > Dear Reviewer L4V3,
> > >
> > > We are truly grateful for your support in moving your rating to accept and for recognizing our efforts to address your concerns about the evaluation and application details. Your thoughtful feedback and constructive comments have been incredibly valuable to us. Thank you again for your time and efforts in reviewing our work!
> > >
> > > Best,
> > >
> > > Paper 9141 Authors

---

### Official Review · Reviewer_rqV8 · 2024-11-04

**Soundness:** 3
**Presentation:** 4
**Contribution:** 2
**Rating:** 8
**Confidence:** 4

**Summary:**

The paper introduces a novel method for intrinsic generation for motion, i.e., augmenting a single input motion sequence into multiple versions. The paper employs an underexplored generative approach - the masked transformer - and presents SOTA performance. To improve diversity and overall quality, the paper also offers a simple KL-divergence based loss to encourage full usage of the latent space, to mitigate mode collapse. The paper reports a complete evaluation, including comparisons, ablation studies, quantitative evaluation, qualitative evaluation and a user study.

**Strengths:**

- Paper is well written and easy to follow.
- Proposed method is simple and elegant
- The mask transformer mechanism is a refreshing approach
- Codebook distribution regularization is interesting
- Evaluation is rigorous, including all conventional metrics

**Weaknesses:**

- The application is a niche one
- The method, being simple, is also limited in contribution
- Qualitative evaluation is limited

**Questions:**

As stated above, I believe that paper presents interesting ideas, and I would like to see this player out there with the other methods.

My main concern is the scope of impact. Perhaps these ideas can and should be examined in a wider scope:
- can the codebook regularization loss improve other VQ methods off the bat?
- More ambitiously, can this method be combined with others, for example can the masked transformer be applied in the bottle neck of a diffusion process? or perhaps instead of the VAE?
- similarly, some more discussion regarding the regularization could benefit the paper. What other alternatives could have been employed?

In addition, I would have liked to see some more qualitative results

Minor concerns:
- Swin is already rather widely adopted and known for shifting windows (Swin-Transformer), I recommend using another name for the sliding window approach (which I find simple and elegant for reducing the receptive field)
- "Crowd motion synthesis" can also be misinterpreted here, as an algorithm to avoid collisions, I would also find a different name here.
- I find figures 1 and 2 a bit exaggerated. The right side of Fig 2 depicts little information for its size. Figure 1 is also rather overwhelming, half of the amount of characters would probably go a long way.

---

> ### Author Response · Authors · 2024-11-22
> **Response to Reviewer rqV8 (Q1, Q2)**
>
> We really appreciate all your encouraging feedback and insightful questions on the scope of impact of our papers. Here are our answers to the first two questions:
>
> ---
> **Q1**: *“Can the codebook regularization loss improve other VQ methods off the bat? ”*
>
> **A1**: Thank you for your thoughtful question about the broader applicability of the codebook regularization loss.  We’ve conducted additional experiments on the impact of this regularization techniques to the tokenization process of other VQ-based methods in motion synthesis, shown in Table 7. and further elaborated in Appendix A.7. We select MMM and MoMask, as they share similar framework with different VQ settings. MMM is backboned on a standard VQ layer with a large codebook, while MoMask leverages residual VQ with multiple codebooks. We apply our codebook regularization loss $\mathcal{L}_{token}$ to the two methods for training the motion tokenization, comparing:
>
> - *R Precision Top 1*: indicates the reconstruction performance or the motion representation capacity of the learned codebook(s).
> - *Perplexity*: measures the codebook utilization.
>
> Table 7 is also shown here:
>
> | **Method** | **R Precision Top 1↑** | **Perplexity↑** |
> | --- | --- | --- |
> | MMM  | 0.503 | 1642.194 |
> | MMM  w/ $\mathcal{L}_{token}$ | **0.572** | **1678.538** |
> | MoMask  | 0.504 | 368.914 |
> | MoMask w/ $\mathcal{L}_{token}$ | **0.504** | **372.702** |
>
> As the results shown in the table, both MMM and MoMask benefit from our proposed $\mathcal{L}_{token}$ with better perplexity, which indicates the overall effectiveness of enhancing codebook utilization. The improvements in both codebook utilization and motion representation capacity are particularly significant for methods like MMM, where the standard VQ layers benefit the most from the regularization. For the more advanced residual VQ architecture used by MoMask, the improvements in codebook utilization are moderate, with no significant gains in motion representation capacity. This may be due to the nature of residual VQ layers, which achieve higher motion representation capacity by utilizing multiple codebooks to quantize the information loss from previous layers rather than directly representing motion data. Consequently, higher codebook utilization alone does not significantly enhance motion representation capacity in this context. Detailed elaboration on the analysis can be found in Appendix A.7.
>
> ---
> **Q2**: “similarly, some more discussion regarding the regularization could benefit the paper. What other alternatives could have been employed?”
>
> **A2**:  Alternative strategies for codebook regularization are listed below:
>
> - Entropy regularization: encouraging entropy maximization of the token distribution can promote codebook utilization, as applied in [3, 4].
> - Stochastic sampling: methods such as those in Zhang et al. (2023) modify sampling strategies to improve token usage across large-scale datasets
>
> The codebook regularization employed in our method specifically addresses the challenge of under-utilization of the discrete latent space in single-instance based motion synthesis. We chose the KL-divergence regularization because of its simplicity and effectiveness in directly addressing the specific challenges of single-instance motion synthesis, as supported by our ablation studies in Table 2.
>
> Due to the current space limited and its relatively lower priority regarding our aimed task, we place the discussion around Q1 and Q2 in Appendix A.7.

---

> ### Author Response · Authors · 2024-11-22
> **Response to Reviewer rqV8 (Q3, Q4)**
>
> Here are the answers to the last two questions:
>
> ---
> **Q3**: “*More ambitiously, can this method be combined with others, for example can the masked transformer be applied in the bottleneck of a diffusion process? or perhaps instead of the VAE?*”
>
> **A3**: We agree that exploring the synergies between generative paradigms is an exciting research direction.
>
> Masked transformers themselves are not suited to serve as bottleneck of diffusion process because they function as standalone generative models, explicitly modelling categorical distributions rather than complementing continuous latent representations. Instead, such integration would possibly be achieved by **applying diffusion process to discretized latent representation**, such as [1] and [2]**.**
>
> For the second part of the question “instead of the VAE”, if the reviewer is referring to **VQVAE + Diffusion model**, we note that while combining VQ tokenization with diffusion processes is possible, current VQVAE approaches often lag behind VAEs in this setting.
>
> ---
>
> **Q4**: *"In addition, I would have liked to see some more qualitative results. Minor concerns:*
>
> - *Swin is already rather widely adopted and known for shifting windows (Swin-Transformer), I recommend using another name for the sliding window approach (which I find simple and elegant for reducing the receptive field)*
> - *"Crowd motion synthesis" can also be misinterpreted here, as an algorithm to avoid collisions, I would also find a different name here.*
> - *I find figures 1 and 2 a bit exaggerated. The right side of Fig 2 depicts little information for its size. Figure 1 is also rather overwhelming, half of the amount of characters would probably go a long way. “*
>
> **A4**: We have updated more visual demonstrations in our updated webpage following your suggestions, including more results on non-human samples and qualitative results of ablation studies. And thank you for pointing out the minor concerns about paper editing. We take some of these suggestions and have completed the revisions in our updated version.
>
> ---
> [1] Chemburkar, A., Lu, S., & Feng, A. (2023). Text-to-Motion Synthesis using Discrete Diffusion Model. *Proceedings of the 34th British Machine Vision Conference (BMVC)*, 624-628.
>
> [2] Kong, H., Gong, K., Lian, D., Mi, M. B., & Wang, X. (2023). Priority-Centric Human Motion Generation in Discrete Latent Space. *Proceedings of the IEEE/CVF International Conference on Computer Vision (ICCV)*, 2023.
>
> [3] Volkov, I. (2022). *Homology-Constrained Vector Quantization Entropy Regularizer*. arXiv preprint arXiv:2211.14363.
>
> [4] Xiao, Y., Shu, K., Zhang, H., Yin, B., Cheang, W. S., Wang, H., & Gao, J. (2024). EGGesture: Entropy-Guided Vector Quantized Variational AutoEncoder for Co-Speech Gesture Generation. *Proceedings of the ACM International Conference on Multimedia (MM)*.
>
> ---
>
> *We sincerely hope that our answers have addressed your concerns, and we’re happy to take any further questions.*

---

> ### Author Response · Authors · 2024-11-25
> **Additional Response to Reviewer rqV8**
>
> Dear Reviewer rqV8,
>
>
> We greatly appreciate your time and efforts reviewing our paper, especially for the positive feedback on our methodology design and your suggestions on exploring further insights about the codebook regularization strategies. We sincerely hope that our responses and updates have fully answered your questions and addressed your concerns. *This is a kind reminder that author-reviewer discussion will be closed in two days.* Please do not hesitate to let us know if you have additional questions. We would be more than happy to open further discussion on them.
>
>
> Best,
>
> Paper 9141 Authors

---

### Author Response · Authors · 2024-11-23
**General Response to All the Reviewers (updated post discussion)**

**We want to first express our big thanks to all reviewers for the thoughtful feedback, as well as for highlighting both the strengths and areas of improvement in our work.** We are particularly grateful for the encouraging comments on our framework, including the novel application of generative masked modeling for single motion synthesis (Reviewers rqV8, Fzyk), the well-motivated and elegantly designed technical contributions such as the codebook regularization loss and sliding window local attention (Reviewers L4V3, oiwR, Fzyk), and the rigorous evaluation through quantitative metrics, qualitative results, and ablation studies (Reviewers rqV8, L4V3, Fzyk).

***During the author-discussion phase***, we truly appreciate the acknowledgement from reviewers for our responses and updates on evaluation metrics consistency (Reviewer L4V3), implementation details for applications (Reviewer L4V3, oiwR), and effectiveness of our proposed sliding window local attention layer (Reviewer Fzyk). ***In our follow-up responses***, we've added additional experiments on latent space visualization as requested (Reviewer oiwR), and further illustrated the novel insights and effectiveness of our key contributions (Reviewer oiwR, Fzyk).

---

In response to the reviewers’ suggestions and comments on further improvements to our work, we have carefully made individual responses to each reviewers correspondingly. To further address the reviewers' concerns, we have also updated our paper and supplementary materials to provide further clarification and demonstrations based on the comments. Below are the key updates:

### **Paper Updates:**

- Renamed **“SwinAttn”** to **“SlidAttn”** and **“crowd motion synthesis”** to **“crowd animation”** for clarity.
- Added clarifications on **Table 1 and Figure 3 consistency** in Section 4.2.
- Expanded **application implementation details** in Appendix A.4, supported by Figure 8.
- Added detailed comparisons of **SlidAttn and QnA layers** in Table 6, supported by analyses in Appendix A.6.
- Included discussions on the **broader impact of codebook regularization** and potential alternatives (e.g., entropy regularization) in Appendix A.7.

### **Visualization Video Demo Updates:**

- Added more results on **non-human skeleton samples** and qualitative ablations in the webpage (Appendix A.1).
- Highlighted **qualitative comparisons of QnA vs. SlidAttn generations** for the newly added ablation study (Appendix A.6 and Table 6).

### **Additional Results in Discussion Post Revision Deadline:**
- Latent space visualization shown and analyzed (see [follow-up response to reviewer oiwR](https://openreview.net/forum?id=d23EVDRJ6g&noteId=MRLUiLpqvu))

---
We would also like to take this opportunity to reiterate the novelty and contributions of MotionDreamer:

1. **Generative Masked Modeling for Single Motion Synthesis:**
MotionDreamer pioneers the application of generative masked modeling for single-instance learning in one-to-many motion synthesis. By modeling explicit categorical distributions of internal motion patterns, it achieves state-of-the-art performance in generating diverse and natural motions that faithfully preserve the reference motion’s internal patterns. *Supporting results: Table 1, Figure 3, and Appendix A.1.*

2. **Codebook Regularization for Single Motion Tokenization:**
To address the inherently imbalanced distribution of internal patterns in single motion sequences, the proposed KL-divergence-based regularization ensures balanced token utilization. This significantly enhances codebook utilization and representation fidelity, enabling robust tokenization for effective single-instance synthesis. *Supporting results: Table 2, Figure 3, Figure 5, and Appendix A.1.*

3. **Sliding Window Local Attention (SlidAttn):**
SlidAttn introduces a novel sliding window computation and aggregation paradigm, which explicitly operate overlapping local windows. This ensures smoother transitions and better correlation across internal motion patterns, which enhances motion fidelity and diversity while addressing overfitting to global patterns common in single-instance setting. *Supporting results: Table 1, Table 6, Appendix A.6, and Appendix A.1.*

4. **Flexible and Lightweight Framework:**
MotionDreamer’s lightweight and adaptable design supports various applications, including crowd animation, temporal editing, and beat-aligned dance synthesis, demonstrating versatility across diverse motion synthesis tasks. *Supporting results: Appendix A.4, Figure 8, and Appendix A.1.*

---

We hope that these updates and additional insights further strengthen our work and demonstrate its novelty and soundness. Thank you again for your valuable feedback and all the efforts.

---

### Author Response · Authors · 2024-12-01

Dear reviewers,

Thank you again for all your time and efforts in reviewing our work! We sincerely appreciate all the insightful comments and suggestions which we believe have helped us improve our work better. So far, we have made our responses to all the initial reviews as well as the follow up comments, and we hope that our responses and updates have answered your questions and addressed your concerns.

*This is a kind reminder that the author-reviewer discussion phase will end in two days.* If our responses and updates have properly addressed your concerns, we would really appreciate it if you could kindly re-evaluate our work. Please do not hesitate to let us know if you have additional questions. We would be more than happy to open further discussion on them.

Best,

Paper 9141 Authors

---

### Author Response · Authors · 2024-12-04
**Update on the general responses with summary post discussion**

Dear reviewers,

We sincerely appreciate your precious time and efforts on the reviews and discussions throughout the weeks. As the end of the rebuttal phase is approaching, we have ***updated our [general response](https://openreview.net/forum?id=d23EVDRJ6g&noteId=uOLyuPigH7)*** by including the summary of author-reviewer discussion, and our highlighted contributions.

Thank you again for all the valuable feedback!

Best,

Paper 9141 Authors

---

### Meta-Review · Area_Chair_Nq38 · 2024-12-19

**Metareview:**

The paper introduces an approach to one-to-many motion synthesis using generative masked modeling. Reviewers generally appreciated the well-designed model, comprehensive evaluations, and practical applications. Strengths include the codebook regularization, sliding window local attention, and strong qualitative results. Some concerns were raised about the technical novelty and comparisons with existing methods like SinMDM. After revisions and clarifications, most concerns were addressed, with ratings improved to acceptance. The AC agreed with Reviewer L4V3 that the framework is well-designed, and its performance is supported by extensive experimental results, although some of the techniques are reminiscent of those applied in other domains. Based on the overall reviews, the paper is recommended for acceptance.

**Additional Comments On Reviewer Discussion:**

Reviewers raised concerns about novelty (oiwR and Fzyk), comparisons with SinMDM (Fzyk), and beat-aligned dance synthesis (L4V3). Authors clarified the novelty of codebook regularization and sliding window attention, provided additional experiments on SinMDM QnA layer, and improved qualitative results. Concerns about evaluation consistency and implementation details were resolved through clarifications and updates. Reviewers oiwR and Fzyk maintained reservations about originality. Given strong evaluations, clear rebuttals, and improved presentation, the AC recommends acceptance. The work would serve as a strong baseline for future work.

---

### Decision · Program_Chairs · 2025-01-22

Accept (Poster)